# How Modular Should Neural Module Networks Be for Systematic Generalization?

**Vanessa D'Amario** [1, 3]  **Tomotake Sasaki** [2, 3]  **Xavier Boix** [1, 3]

[1] Massachusetts Institute of Technology, USA  [2] Fujitsu Limited, Japan
[3] Center for Brains, Minds and Machines, USA
`vanessad@mit.edu, tomotake.sasaki@fujitsu.com, xboix@mit.edu`

## Abstract

Neural Module Networks (NMNs) aim at Visual Question Answering (VQA) via composition of modules that tackle a sub-task. NMNs are a promising strategy to achieve systematic generalization, *ie.* overcoming biasing factors in the training distribution. However, the aspects of NMNs that facilitate systematic generalization are not fully understood. In this paper, we demonstrate that the degree of modularity of the NMN have large influence on systematic generalization. In a series of experiments on three VQA datasets (VQA-MNIST, SQOOP, and CLEVR-CoGenT), our results reveal that tuning the degree of modularity, especially at the image encoder stage, reaches substantially higher systematic generalization. These findings lead to new NMN architectures that outperform previous ones in terms of systematic generalization.

## 1 Introduction

The combinatorial nature of vision [1] is an open challenge for learning machines tackling Visual Question Answering (VQA) [2–7]. The amount of possible combinations of object categories, attributes, relations, context and visual tasks is exponentially large, and any training set contains a limited amount of such combinations. The ability to generalize to combinations not included in the training set, *ie.* the so-called *systematic generalization* [8–10], is a hallmark of intelligence.

Recent research has shown that Neural Module Networks (NMNs) are a promising approach for systematic generalization in VQA [5, 11]. NMNs infer a program layout containing a sequence of sub-tasks, where each sub-task is associated to a module consisting of a neural network [12, 13].

The aspects that facilitate systematic generalization in NMNs are not fully understood. Recent works demonstrate that modular approaches are helpful for object recognition tasks with novel combinations of category and viewpoints [14] and also novel combinations of visual attributes [15]. Bahdanau *et al.* [11] have demonstrated the importance of using an appropriate program layout for systematic generalization, and Bahdanau *et al.* [5] in another paper have shown that the module's network architecture also impacts systematic generalization. These results highlight that using modules to tackle sub-tasks and using compositions of these modules facilitates systematic generalization. However, there is a central question at the heart of modular approaches that remains largely unaddressed: what sub-tasks should be tackled by a module? Namely, is it best for systematic generalization to use a large degree of modularity such that modules tackle very specific sub-tasks, or is it best to use a smaller degree of modularity?

In this paper, we investigate what sub-tasks a module should perform in order to facilitate systematic generalization in VQA. We analyse different degrees of modularity as in the example of Figure 1, where NMNs could have three different degrees of modularity: (i) a unique module for tackling all

35th Conference on Neural Information Processing Systems (NeurIPS 2021).

sub-tasks, (ii) modules tackling groups of sub-tasks, or (iii) many modules each one for tackling sub-tasks at the highest degree of modularity. Also, we investigate the degree of modularity at all the stages of the network, as NMNs have a structure divided in three stages: an *image encoder* for the extraction of visual features, a set of *intermediate modules* arranged based on the program layout, and a *classifier* that provides the output answer.

We examined the degree of modularity in three families of VQA datasets: VQA-MNIST, the SQOOP dataset [11], and the Compositional Generalization Test (CoGenT) of CLEVR [2]. Our results reveal two main findings that are consistent across the tested datasets: (i) an intermediate degree of modularity by grouping sub-tasks leads to much higher systematic generalization than for the NMNs' modules introduced in previous works, and (ii) modularity is mostly effective when is defined at the image encoder stage, which is rarely done in the literature. Furthermore, we show that these findings are directly applicable to improve the systematic generalization of state-of-the-art NMNs.

## 2 Libraries of Modules

NMNs consist of three components: a program generator, a library of modules and an execution engine. The program generator translates VQA questions into program layouts, consisting of compositions of sub-tasks. The library of modules contains neural networks that perform sub-tasks, and these modules are used by the execution engine to put the program layout into action.

Given that the use of a correct program layout plays a fundamental role to achieve systematic generalization with NMNs [11, 13], we follow the approach in previous works that fix the program layout to the ground-truth program, such that it does not interfere in the analysis of the library [5, 11]. Thus, we follow a reductionist approach, which facilitates gaining an understanding of the crucial aspects of NMNs in systematic generalization. As we gain such understanding, we will be in a position to study the interactions between these aspects in future works.

Here, we aim at understanding whether the choice of a library with a specific degree of modularity facilitates systematic generalization in NMNs. Figure 1a shows examples of several libraries of modules (one library per column), characterized by different degrees of modularity at different stages of the network. Each row identifies a stage in the network, while the use of single or multiple modules identifies the degree of modularity at each stage. In particular, a library with a single module shared across all sub-tasks is called *all*. A library with modules tackling groups of sub-tasks is called *group*. Finally, a library with modules that tackle each single sub-task is called *sub-task*. These different ways of defining the degree of modularity in the library are detailed in Figure 1b.

Note that in the intermediate stage, it is sometimes necessary to use modules that operate differently depending on an input argument related to the sub-task to be performed by the module. For example, a module whose sub-task is to determine whether an object is of a specific color or not, could be implemented by a module specialized to detect colors with an input argument that indicates the color to be detected. It could also be implemented by a module specialized to detect a specific color, which would not require an additional input argument.

In Figure 1c, we provide an example of usage of the libraries given the question *"Is the green object left of '2'?"*. The ground-truth program layout is a tree structure with two leaves and a root that identify respectively the pair of objects and the spatial relation. In the figure we depict how the same program layout can be implemented with libraries with different degrees of modularity. Note that the degree of modularity does not change the final number of modules in the program layout, but the number of modules in the library.

In the following, we first introduce the definitions of the modules in the library for the image encoder and the classifier stages, and then we introduce existing implementations of intermediate modules and propose a new one.

### 2.1 Libraries of Modules at the Image Encoder and Classifier Stages

The image encoder is usually a single convolutional neural network module common to all sub-tasks (first two columns of Figure 1a). We analyze image encoders defined per group of sub-tasks, as in the third and forth columns in Figure 1a. The library with maximum degree of modularity at the image

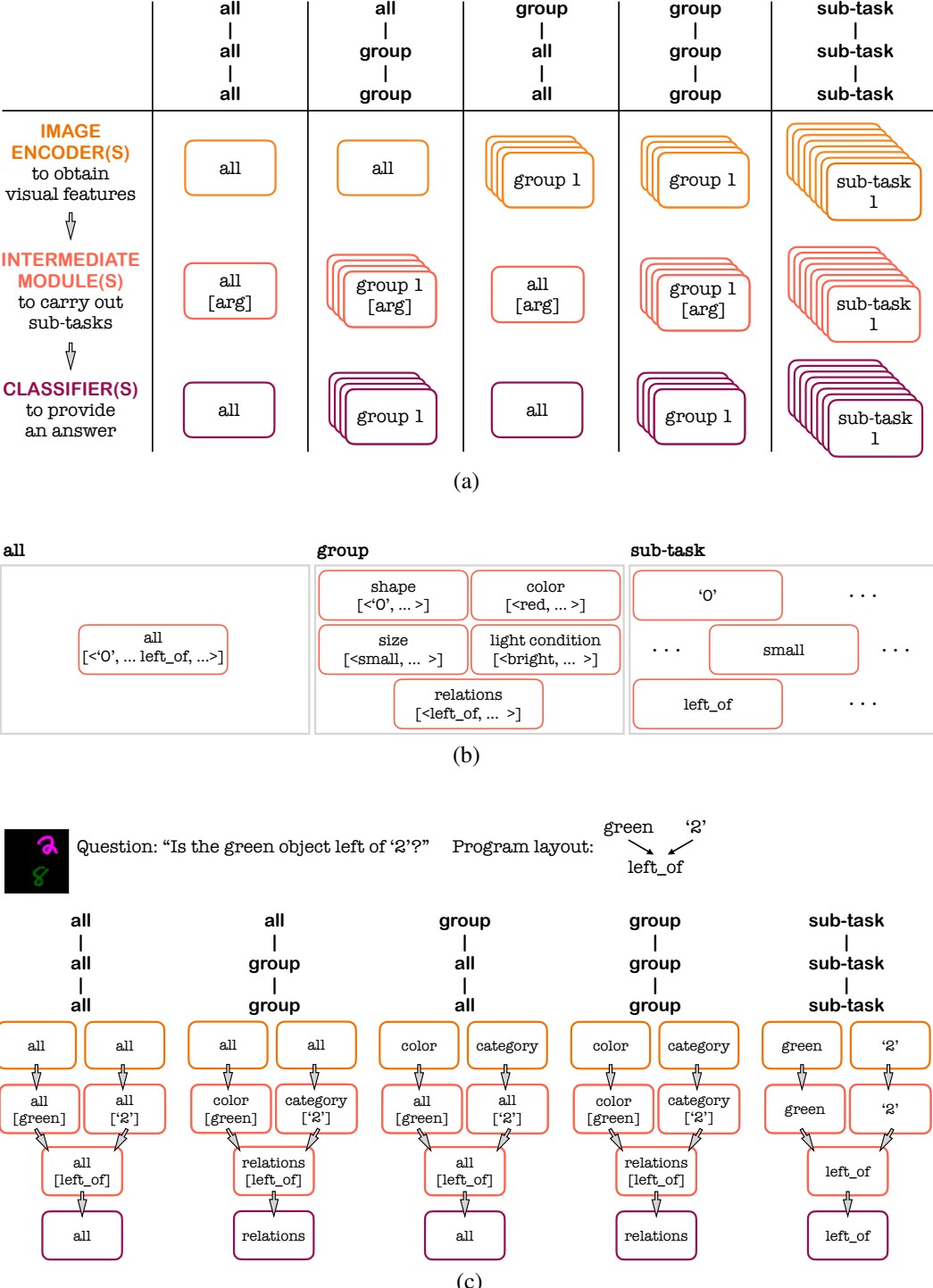

Figure 1: *Libraries of modules.* (a) Example of five libraries with different degrees of modularity (*all*, *group*, and *sub-task*) analyzed in this paper. Some modules have additional input arguments to indicate specifics of the sub-task. (b) Intermediate modules from libraries with *all*, *group*, and *sub-task* degrees of modularity, for the VQA-MNIST dataset. (c) Composition of modules leveraging on the libraries in (a), given the question *"Is the green object left of '2'?"* and an image containing a pair of objects (the former identified by its color, the latter by its category).

encoder stage corresponds to having an image encoder per sub-task of the VQA problem, as depicted in the last column of Figure 1a.

For a VQA task with binary answers, libraries with different degrees of modularity can be equally defined on the classifier.

## 2.2 Libraries of Modules at the Intermediate Modules Stage

The architecture of the intermediate modules varies across the literature. We consider two architectures which previous works [5, 11] have focused on: the *Find* [16] and the *Residual* [13] modules. Let $f(k, s_x, s_y)$ be the function that represents an intermediate module, where each sub-task is denoted by an index $k$ that represents the input argument of the module, and $(s_x, s_y)$ are the data inputs to the module, which can be the representation of the input image or the output of the precedent intermediate modules according to the program layout. If a module requires a single input for the given sub-task, $s_y$ is the output of the image encoder given a null input image.

In the following, we show that the *Find module* can be interpreted as a library with a *single intermediate module* shared across all sub-tasks, with the neural representation modulated by an embedding related to the sub-task. Then, we show that the *Residual module* corresponds to a library with maximum degree of modularity, in which each sub-task corresponds a single intermediate module. Finally, we introduce a new module based on these two, which has an intermediate degree of modularity.

**A single intermediate module (aka. Find module [16]).**    The definition is the following:

$$\gamma_k = \text{Embedding}(k), \tag{1}$$

$$f(k, s_x, s_y) = \text{ReLU}(W_1 * (\gamma_k \odot \text{ReLU}(W_2 * [s_x; s_y] + b_2)) + b_1), \tag{2}$$

where $[s_x; s_y]$ is the concatenation of the two inputs and $[W_1; b_1; W_2; b_2]$ are the weights and bias terms of the convolutional layer. These weights and bias terms are the same across all sub-tasks, *ie.* all the sub-tasks are tackled with the same module. The module performs a sub-task by "modulating" the neural activity via the element-wise product involving $\gamma_k$. Figure 1b depicts the Find module at the first column.

**One module per sub-task (aka. Residual module [13]).**    This represents a library with a much finer degree of modularity than the previous one. For each sub-task $k$, we have an independent set of convolutional weights indexed by $k$, *ie.* $[W_1^k; b_1^k; W_2^k; b_2^k; W_3^k; b_3^k]$, as in the following:

$$\tilde{s}_k = \text{ReLU}(W_3^k * [s_x; s_y] + b_3^k), \tag{3}$$

$$f(k, s_x, s_y) = \text{ReLU}(\tilde{s}_k + W_1^k * \text{ReLU}(W_2^k * \tilde{s}_k + b_2^k) + b_1^k). \tag{4}$$

Note that here the mechanism to tackle a sub-task is to use an independent set of weights per sub-task, as depicted in Figure 1b, at the third column.

**One module per group of sub-tasks.**    We introduce a new library of modules that has a degree of modularity halfway between the two presented above (depicted in Figure 1b, second column). Each modules tackles a group of sub-tasks via a Find module. We use $[W_1^g; b_1^g; W_2^g; b_2^g]$ to denote the weights and bias terms of a module for the group of sub-tasks indexed by $g$. Let $g = \text{Group}(k)$ be a mapping from the sub-task $k$ to the index of the group, $g$. Thus, this library of modules is defined as follows:

$$g = \text{Group}(k), \quad \gamma_k = \text{Embedding}(k), \tag{5}$$

$$f(k, s_x, s_y) = \text{ReLU}(W_1^g * (\gamma_k \odot \text{ReLU}(W_2^g * [s_x, s_y] + b_2^g)) + b_1^g). \tag{6}$$

Note that this library of modules uses both Find and Residual architectures to tackle a sub-task—the division of sub-tasks in groups of separated weights reflects the design of the Residual architecture, while the modulation mechanism that allows the modules to adjust their representation among sub-tasks in the same group reflects the design of the Find architecture.

## 3   Datasets for Analysing Systematic Generalization in VQA

In this section, we introduce the VQA-MNIST and the SQOOP datasets to study the effect of libraries with different degrees of modularity on systematic generalization.

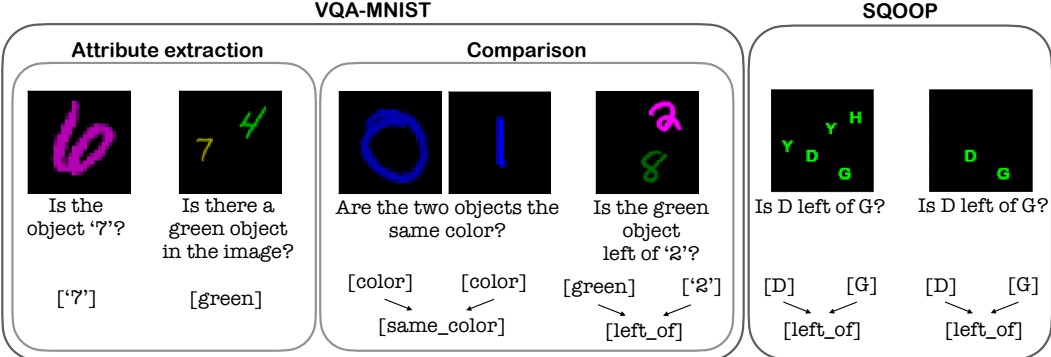

Figure 2: *VQA datasets to evaluate systematic generalization.* From top to bottom: Visual input, question and corresponding program layout. The VQA-MNIST contains objects that are combinations of four attributes (category, color, size and illumination condition). The tasks consist on attribute extraction (from a single object and from multiple objects), and attribute comparisons (between separated object pairs and spatial locations). The training distribution has a limited amount of attribute combinations. The SQOOP dataset contains objects at different positions and the task consists on comparing the position of two given objects. The training distribution has a limited number of co-occurring objects in the image. The original SQOOP dataset contains five objects per image [11], and we further limit the training distribution by only allowing two objects per image.

## 3.1 VQA-MNIST: limited combinations of visual attributes

We introduce VQA-MNIST, a family of four VQA datasets of which two of them are related to attribute extraction, while the other two deal with comparisons of attributes and spatial positions. All tasks require a binary answer ('yes' or 'no'). Each object is an MNIST digit [17] with multiple attributes: category, color, size and illumination condition among respectively ten, five, three, and three different options. This leads to a total amount of attribute instances equal to $21$, and a total amount of combinations of attribute instances equal to $450$. See Appendix A.1 for more details.

To generate a limited set of combinations of attributes, we fix a parameter $r$ and we randomly extract combinations among all the $450$ possibilities, with the requirement that each of the $21$ attribute instances appears at least $r$ times (see Appendix A.2 for details). In order to study the effect of training data diversity, we compare NMNs trained with datasets with different number of attribute combinations and for a fair comparison, we evaluate them under the same testing conditions, as described next. We define *data-run* as six different training datasets originated from training combinations obtained with six different $r$'s ($[1, 2, 5, 8, 10, 20]$), such that the set of training combinations extracted for smaller values of $r$ is included in the training combinations for larger values of $r$. We evaluate systematic generalization for all datasets in the data-run on the same images that come from novel combinations, obtained by fixing $r = 5$. We generate the training and test examples as described in Appendix A.3. The amount of training examples for each dataset in the data-run is fixed to 210K for training, and 42K for testing systematic generalization. In this way, all NMNs in the data-run are trained with the same amount of data, *ie.* we evaluate different amounts of training data diversity for the same number of training images.

In Figure 2, we depict the four VQA tasks divided into *attribute extraction* and *attribute comparison*, which are summarized in the following:

• *Attribute extraction.* Starting from the left of Figure 2, we show two datasets, *attribute extraction from single object* and *attribute extraction from multiple objects*. In the former case, the image contains a single object (image size is $28 \times 28$ pixels), in the latter, it always contains a pair of objects (image size is $64 \times 64$ pixels). The second object plays the role of a confounding factor. In both datasets, the questions inquire about one of the 21 possible attribute instances. The ground-truth program layout corresponds to a single sub-task, which is identified by the attribute instance in the question. Thus, the program layout in the intermediate module corresponds to a single module identifying the sub-task (the sub-tasks in Figure 2 are identified by *'7'* and *green*).

- *Attribute comparison.* At the center of Figure 2 we show two datasets for attribute comparison. These have a ground-truth program layout that is a tree structure. The first dataset consists on *attribute comparison between pairs of separated objects*, in which comparisons are done between a pair of objects contained in separated images (of size $28 \times 28$ pixels). The comparison are with respect to an attribute of the dataset. The two leaves in the program layout extract the attribute and the root tackles the comparison between them. The second dataset consists on a *comparison between spatial positions*, as shown in Figure 2 (image size $64 \times 64$ pixels). The leaves of the program layout identify the objects in the image, and the root tackles the spatial comparison (*ie.* above, below, left of, or right of).

## 3.2 SQOOP: limited amount of co-occurrence of objects in the images

The SQOOP dataset tackles the problem of spatial relations between two objects [11]. Namely, the questions consist on comparing the position of two objects, *e.g., Is D left of G?* The dataset contains four spatial relations (above, below, left of, or right of) on thirty-six possible objects. All objects can appear in the image, but specific pairs of objects are not compared during training. By limiting the possible pairs of objects compared during training, systematic generalization is measured as the ability to compare novel pairs of objects.

In the original SQOOP dataset [11], each image contains five objects, as in the left example of the SQOOP block in Figure 2. We increase the bias of the SQOOP dataset, to make it more challenging for NMNs. To do so, we reduce the objects in the image to the pair in the question. We generate a dataset with the smallest amount of objects co-occurring with another object in an image, such that each object only co-occurs with another object. Given a generic training question *e.g., "Is D left of G?"*, all the training images related to this question contain the *D* and *G* objects only. This procedure substantially reduces the training data diversity of the original SQOOP datasets with five objects per image.

# 4 Results

In this section, we report the systematic generalization accuracy of NMNs with different libraries of modules on the VQA-MNIST and the SQOOP datasets.

## 4.1 VQA-MNIST

To analyze the impact of the degree of modularity of the library in systematic generalization, we report results using the following libraries, which are depicted in Figure 1a:

- *all - all - all:* This is a library with a single image encoder module, a single classifier module, and intermediate module defined as the Find module. This library is commonly used in NMN [16], while the following ones are explored for the first time in this paper.

- *all - group - group:* This library contains a single image encoder, and intermediate modules and classifiers separated per group. We divide sub-tasks related to object categories, colors, spatial relations and so forth in different groups.

- *group - all - all:* This is a library with multiple image encoders separated per group, where sub-tasks related to object categories, colors, spatial relations and so forth belong to different groups. It has a single intermediate module and a single classifier shared across all sub-tasks, respectively.

- *group - group - group:* This library has one image encoder module per each group of sub-tasks, as in the image encoder of *group - all - all*. The intermediate modules and the classifier also tackle groups of sub-task, as in the intermediate modules and classifier of *all - group - group*.

- *sub-task - sub-task - sub-task:* This is the library with the maximum degree of modularity. The modules address a single sub-task at each stage of the NMN.

Other libraries of modules are analyzed in Appendix B, such as other state-of-the-art NMN implementations and different libraries that help to isolate the effect of the degree of modularity at the different stages. The analysis of the libraries in the Appendix serves to strengthen the evidence that support the conclusions obtained with the libraries analysed in this section.

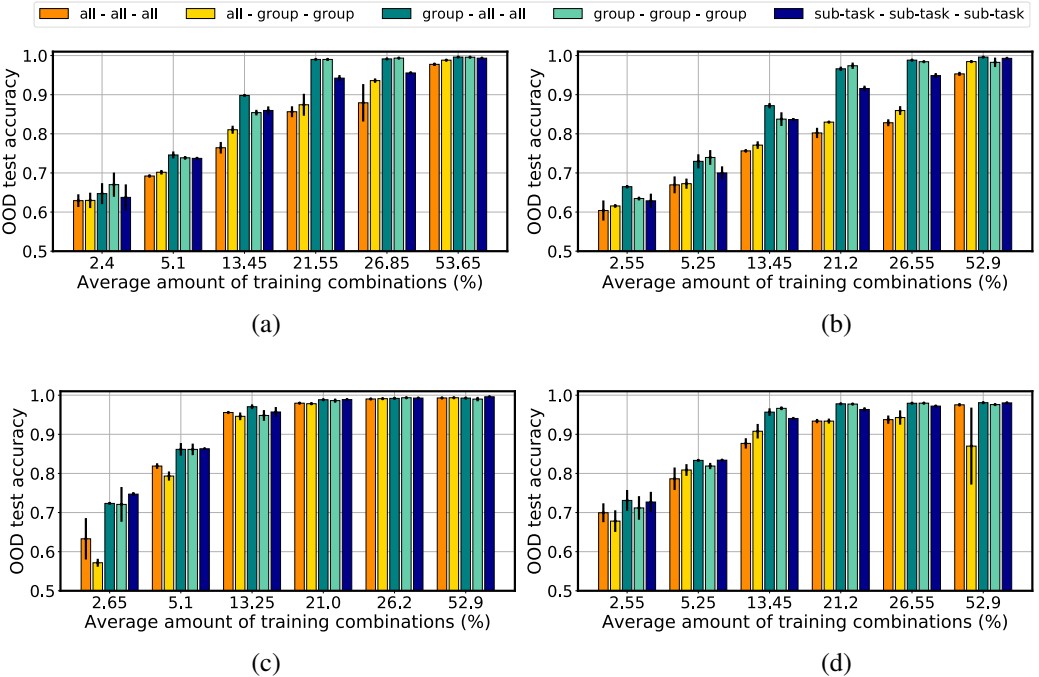

Figure 3: *Results in VQA-MNIST.* Systematic generalization accuracy (referred as OOD test accuracy in the plot) that evaluates VQA on novel combinations of visual attributes, for different amount of training combinations (%), *ie.* training data diversity. Results compare different libraries of modules in the four types of tasks of VQA-MNIST: (a) attribute extraction from single object, (b) attribute extraction from multiple objects, (c) attribute comparison between pairs of separated objects, (d) comparison between spatial positions.

To measure systematic generalization accuracy, it needs to be guaranteed that the novel combinations of attributes used for testing are not used in any stage of learning [6]. We perform a grid-search over the NMNs' hyper-parameters, and evaluate the accuracy using an in-distribution validation split at the last iteration of the training. The model with highest in-distribution validation accuracy is selected, such that the novel combinations of attributes are not used for hyper-parameter tuning (see Appendix B.2 for more details).

Figure 3 shows the systematic generalization accuracy to novel combinations of attributes on the four tasks in VQA-MNIST. Across all plots, each value on the horizontal-axis displays the average percentage of training combinations for a fixed $r$ (recall that $r$ is the minimum amount of times an attribute instance appears in the training combinations) across two data-runs, where the bars are grouped based on the value of $r$. In the legend, we specify the library of modules used in the NMN. On the vertical-axis, we report mean and standard deviation of the systematic generalization accuracy across the two data-runs.

**Systematic generalization compared to in-distribution generalization is much more challenging.** Even though NMNs achieve the highest systematic generalization in the literature [5, 11], Figure 3 shows that the systematic generalization accuracy across VQA tasks can be very low when the amount of training combinations is small. This highlights the difficulty to generalize to novel combinations of attributes when the training datasets are biased. Recall that VQA-MNIST is a synthetic dataset which could be considered one of the simplest. Yet, state-of-the-art approaches in systematic generalization have a remarkable low accuracy in this dataset. Appendix B.3 shows the in-distribution accuracy, which is in stark contrast with the systematic generalization accuracy, as it is close to 100% accuracy in all evaluated cases. We underline that the low systematic generalization accuracy depends on the amount of training combinations, *ie.* the training data diversity, which has been reported in other contexts beyond VQA, such as in object recognition [14, 15] and in navigation tasks [8, 18].

Table 1: *Results in the SQOOP dataset.* Systematic generalization accuracy (%) for NMNs with tree program layout trained on SQOOP. Top row: SQOOP dataset with five objects per image, bottom row: SQOOP dataset with two objects per image. The *all - sub-task - all* library reaches higher systematic generalization in the more difficult case of two objects per image.

| | *all - all - all* | *all - sub-task - all* |
|---|---|---|
| Five objects per image (as in previous work [11]) | **99.8 ± 0.2** | **99.96 ± 0.06** |
| Two objects per image | 84 ± 2 | **88.5 ± 0.5** |

**Libraries with an intermediate degree of modularity, especially in the image encoder stage, substantially improve systematic generalization.** The *group - all - all* and the *group - group - group* libraries achieve the highest systematic generalization accuracy across all VQA tasks. These libraries also clearly outperform the *sub-task - sub-task - sub-task* library, which has the highest degree of modularity. The library with worst systematic generalization is *all - all - all*, which has the lowest amount of modularity. Observe that the improvements of systematic generalization accuracy are large in some cases (there are gaps of about 15% accuracy between libraries of modules). Also, note that the *group - all - all* library is clearly superior to *all - group - group*, which suggests that a higher degree of modularity is more helpful at the image encoder stage. Taking these results together, we can conclude that a library with an intermediate degree of modularity (*group*), specially at the image encoder stage, is crucially important to improve systematic generalization for novel combinations of visual attributes. In Appendix B, we report several control experiments that support this conclusion. Namely, in Appendix B.4 we validate the conclusions with other libraries of modules commonly used in the literature (such as *all - sub-task - all*, and variants with batch normalization applied to the *all* libraries, which is when it is only possible). In Appendix B.5, B.6 and B.7 we further validate the conclusion analysing variants for the image encoder, intermediate modules and image classifier, respectively.

**Higher systematic generalization is not reachable with larger amounts of training examples.** Since using libraries with a larger number of modules may lead to modules being trained with lower number of training examples, we controlled that our conclusions are not affected by increasing the amount of training examples. We consider the task of attribute extraction from single object and attribute comparison between objects in separated images, and increased the training set size of a factor ten (for a total of 2.1M training examples), while keeping the same percentage of training combinations of the previous experiments in Figure 3 (a,c). Results are reported in Appendix B.8, which show that the systematic generalization accuracy is not improved when increasing the number of training examples, and conclusions are consistent to those in Figure 3 (a,c). Thus, the amount of training examples in these experiments already achieves the highest possible systematic generalization accuracy by increasing the dataset size, *ie.* our conclusions are not a consequence of having libraries with data-starved modules.

### 4.2 SQOOP

We use the two libraries of modules used in previous work [11]: *all - all - all* and *all - sub-task - all*. To measure systematic generalization, we consider the model at its last training iteration using the hyper-parameters in previous works (batch normalization is used in the image encoder and classifier stages).

In Table 1, we show the systematic generalization accuracy (mean and standard deviation across five trials). Results are consistent with the conclusions obtained in VQA-MNIST. On the original SQOOP dataset with five objects per image, both libraries achieve almost perfect systematic generalization over novel test questions. For the dataset with two objects in the image, the dataset is more biased because of the lower amount of co-occurring objects, and hence, it increases the difficulty of generalizing to novel question. In this case, the *all - sub-task - all* library achieves significantly higher systematic generalization performance than the *all - all - all* library. Finally, in Appendix C we report the performance of other VQA models that do not use modules (*ie.* FiLM [19] and MAC [11, 20]). The results provide re-assurance that NMNs achieve higher systematic generalization.

Table 2: *Results in the CLEVR-CoGenT dataset.* Mean and standard deviation of test accuracy (%) across five repetitions. NMNs are tested on in-distribution and out-of-distribution (systematic generalization).

|  | Tensor-NMN with *all* image encoder | Vector-NMN with *all* image encoder | Vector-NMN with *group* image encoder (**ours**) |
| --- | --- | --- | --- |
| in-distribution | $97.9 \pm 0.1$ | $\mathbf{98.0 \pm 0.2}$ | $94.4 \pm 0.3$ |
| syst. generalization | $72.7 \pm 0.5$ | $73.2 \pm 0.2$ | $\mathbf{77.3 \pm 1.3}$ |

## 5   Application on CLEVR-CoGenT Split for Systematic Generalization

In this section, we apply the findings in the previous section on existing NMN architectures used in practice. To do so, we use CLEVR, a diagnostic dataset in VQA [2]. This dataset consists of complex 3D scenes with multiple objects and ground-truth program layouts formed by compositions of sub-tasks. This dataset comes with additional splits to test systematic generalization, namely the Compositional Generalization Test (CLEVR-CoGenT). CLEVR-CoGenT is divided in two conditions where cubes and cylinders appear in a limited amount of colors, that are inverted between training and testing (see Appendix D.1 for details). In this way, we can measure systematic generalization to novel combinations of visual attributes (shape and color in this case).

We now show that the analysis in previous sections is helpful to improve state-of-the-art NMNs. We focus on Vector-NMN, which recently has been shown to outperform all previous NMNs [5]. Vector-NMN uses a single image encoder shared across all sub-tasks. The Vector-NMN's image encoder takes as input the features extracted from a pre-trained ResNet-101 [21], and consists of two trainable convolutional layers shared among all sub-tasks. The details of Vector-NMN definition can be found in Appendix D.2. Our variant of Vector-NMN is based on tuning the degree of modularity in the encoder stage, as we have previously shown this can substantially improve the systematic generalization capabilities. Thus, we modify the library of Vector-NMN by using a higher degree of modularity at the image encoder stage, *ie.* one module for each group of sub-tasks. Concretely, we define the groups as: (i) counting tasks, (ii) tasks related to colors, (iii) materials, (iv) shapes, (v) sizes, (vi) spatial relations, (vii) logical operations, (viii) input scene (for details, see Appendix D.3).

The hyper-parameters are fixed to the ones in the previous work [5], and the ground-truth program layout is given. Tables 2 and 3 show the mean and standard deviation accuracy (across five runs with different network initialization each) of the original Vector-NMN, our version of Vector-NMN with a more modular (*group*) image encoder, and also Tensor-NMN, which is another NMN commonly used in the literature that here serves as baseline. Table 2 shows the in-distribution accuracy and the systematic generalization accuracy in objects with novel combinations of shape and color. Results shows the wide gap between in-distribution and out-of-distribution for all NMNs. Our Vector-NMN with a more modular (*group*) image encoder achieves the best systematic generalization performance. This comes with a reduction of in-distribution generalization, which can be explained by the trade-off between in-distribution and out-of-distribution generalization reported in previous works [14, 22].

Table 3 shows the break-down of systematic generalization for each type of question. Note that the systematic generalization accuracy varies depending on questions types. The questions that are mostly affected by the limited amount of object shapes and color combinations in CLEVR-CoGenT, *ie.* query_shape and query_color, is where a higher degree of modularity in the image encoder brings an improvement of the accuracy of 12% and 7%, respectively. A broader comparison of our approach with other non-modular approaches such as FiLM [19] and MAC [5, 20], and other NMN variants is reported in Appendix D.4. These results further demonstrate the higher systematic generalization accuracy of our approach.

## 6   Conclusions and Discussion

Our results demonstrate that NMNs with a library of modules with an intermediate degree of modularity, especially at the image encoder stage, substantially improves systematic generalization. This finding is consistent across datasets of different complexity and NMN architectures, and it is easily applicable to improve state-of-the-art NMNs by using modular image encoders.

Table 3: *Breakdown of the results in the CLEVR-CoGenT dataset.* Breakdown by question type. Systematic generalization accuracy (%) is reported, and it is the average across five trials.

| | Tensor-NMN with *all* image encoder | Vector-NMN with *all* image encoder | Vector-NMN with *group* image encoder (**ours**) |
|---|---|---|---|
| count | $69.7 \pm 0.8$ | $70.4 \pm 0.4$ | $\mathbf{71 \pm 1}$ |
| equal_color | $75.6 \pm 0.8$ | $74 \pm 1$ | $\mathbf{80 \pm 1}$ |
| equal_integer | $82.7 \pm 0.3$ | $78 \pm 2$ | $\mathbf{85 \pm 2}$ |
| equal_material | $74 \pm 2$ | $74.2 \pm 0.7$ | $\mathbf{84 \pm 2}$ |
| equal_shape | $\mathbf{91 \pm 2}$ | $89 \pm 3$ | $79 \pm 2$ |
| equal_size | $75 \pm 1$ | $75 \pm 1$ | $\mathbf{88 \pm 2}$ |
| exist | $84.2 \pm 0.4$ | $\mathbf{84.4 \pm 0.4}$ | $84.4 \pm 0.5$ |
| greater_than | $83.8 \pm 0.6$ | $83.6 \pm 0.4$ | $\mathbf{89 \pm 1}$ |
| less_than | $80.7 \pm 0.9$ | $82.0 \pm 0.5$ | $\mathbf{87 \pm 2}$ |
| query_color | $58 \pm 1$ | $60 \pm 1$ | $\mathbf{67 \pm 4}$ |
| query_material | $84.1 \pm 0.9$ | $84.7 \pm 0.4$ | $\mathbf{88.2 \pm 0.8}$ |
| query_shape | $37 \pm 1$ | $40 \pm 3$ | $\mathbf{52 \pm 3}$ |
| query_size | $83.5 \pm 0.6$ | $84.7 \pm 0.7$ | $\mathbf{89.5 \pm 0.5}$ |

This work also has led to new research questions. While we have shown that modularity has a large impact in systematic generalization, we have tuned the degree of modularity at hand and in practical applications it is desirable to do this automatically. Also, it is unclear how other types of bias, such as bias in the program layout, could affect systematic generalization. To motivate follow-up work in other types of bias beyond the ones analyzed in this work, in Appendix E we report a pilot experiment that shows that bias in the program layout could degrade systematic generalization of the NMNs we introduced. This suggests that there may be a trade-off between the degree of modularity and bias in the program layout, which will be investigated in future works.

Finally, we are intrigued about the neural mechanisms that facilitate systematic generalization and how the degree of modularity affects those mechanisms. Some hints towards an answer have been pointed out by [14], which shows that the emergence of invariance to nuisance factors at the individual neuron level improves systematic generalization for object recognition.

## Acknowledgments and Disclosure of Funding

We would like to thank Moyuru Yamada, Hisanao Akima, Pawan Sinha and Tomaso Poggio for useful discussions and insightful advice provided during this project, and Ramdas Pillai and Atsushi Kajita from Fixstars Solutions for their assistance executing the large scale computational experiments. This work has been supported by the Center for Brains, Minds and Machines (funded by NSF STC award CCF-1231216), the R01EY020517 grant from the National Eye Institute (NIH) and Fujitsu Limited (Contract No. 40008819 and 40009105).

## Code and Data Availability

Code and data can be found in the following github repository:
https://github.com/vanessadamario/understanding_reasoning.git.

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
