# A  VQA-MNIST Dataset

In Appendix A.1, we describe the attributes and the comparisons in the VQA-MNIST datasets. The algorithm for the generation of the training and testing combinations is in Appendix A.2. Given the combinations, in Appendix A.3 we define the procedure to generate training and testing examples.

## A.1  Attributes and Comparisons

**Visual attributes.**   Each object in VQA-MNIST is characterized by a category, the original MNIST category from 0 to 9, a color among red, green, blue, yellow and pink, a light condition among bright, half-bright, and dark, correspondent to a multiplicative factor $(1, 0.7, 0.4)$ applied on the whole image, and a size among large, medium, small, correspondent to three scale factors $(1, 5/7, 1/2)$ of the original digit. This corresponds to a total of 21 attribute instances. Figure App.1 shows some objects with a reduced amount of combination of attributes.

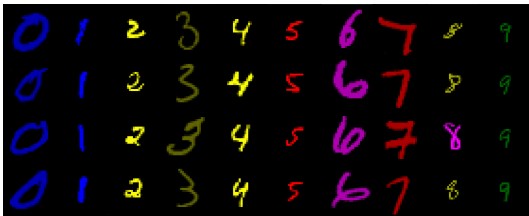

Figure App.1: VQA-MNIST: some examples of training images, with a limited amount of attributes combinations.

**Attribute comparisons.**   The sub-tasks used for attribute comparison between two separated objects are in Table App.1. The sub-tasks for comparison of spatial relations are in Table App.2.

Table App.1: Sub-tasks for attribute comparison between pairs of separated objects. The sub-tasks are divided in groups.

| group of sub-tasks | sub-tasks |
|---|---|
| category | `category` |
| color | `color` |
| light | `light` |
| size | `size` |
| comparison of category | `same_category`; `different_category` |
| comparison of color | `same_color`; `different_color` |
| comparison of light condition | `same_light`; `different_light`; `brighter`; `darker` |
| comparison of size | `same_size`; `different_size`; `larger`; `smaller` |

Table App.2: Sub-tasks for comparison of spatial positions between two objects. The sub-tasks are divided in groups.

| group of sub-tasks | sub-tasks |
|---|---|
| category | 0; 1; 2; 3; 4; 5; 6; 7; 8; 9 |
| color | `red`; `green`; `blue`; `yellow`; `pink` |
| light | `bright`; `half-bright`; `dark` |
| size | `large`; `medium`; `small` |
| spatial relations | `left_of`; `right_of`; `above`; `below` |

## A.2  Algorithm for the Generation of Training and Test Combinations

To generate the training and test combinations across all the VQA-MNIST datasets, we fix the set of attribute types: category (attr1), color (attr2), light condition (attr3), and size (attr4);

and we define a list of attribute instances for each type. The list $\text{attr1} = [0, \ldots, 9]$ contains the ten categories, the list $\text{attr2} = [\texttt{red, green, blue, yellow, pink}]$ the five colors, the list $\text{attr3} = [\texttt{bright, half-bright, dark}]$ the three light conditions, and the list $\text{attr4} = [\texttt{small, medium, large}]$ the three sizes. We define a single comprehensive list of all attributes as $\text{attributes} = \text{attr1} + \text{attr2} + \text{attr3} + \text{attr4}$, of length 21. Two integers are required for the generation of training and test combinations: n_combinations_train and n_combinations_test, which define the minimum amount of times (indicated with the $r$ parameter) each attribute will appear in the training and testing sets, respectively. We also define train_combinations and test_combinations as two empty lists. Through a stochastic procedure described in Algorithm 1, we fill these lists with the objects for the training and the testing sets.

---

**Algorithm 1** Generation of Training and Test Combinations

---

**Require:** n_train_combinations, n_test_combinations
1: train_combinations, test_combinations $= [\,], [\,], k = 0$, random_seed $= n$
2: **for** $k <$ n_test_combinations **do**
3:
4:      taken $= \text{dict}()$
5:      **for** $a\_$ in attributes **do**
6:          taken$[a\_] = False$
7:      **end for**
                 ▷ At every iteration in the loop we set the dictionary of seen attribute to False
8:
9:      missing_attributes $= True$
10:      **while** missing_attributes **do**
                         ▷ Until we see all the attributes at the $k$-th iteration
11:          candidate $= [\text{choice}(\text{attr1}), \text{choice}(\text{attr2}), \text{choice}(\text{attr3}), \text{choice}(\text{attr4})]$
                         ▷ random extraction using uniform distribution
12:          exists $=$ candidate in test_combinations
13:          reject $= True$        ▷ We assume that all attributes in candidate already appeared
14:          **for** $c\_$ in candidate **do**
15:              reject $\ast =$ taken$[c\_]$
16:          **end for**
17:          **if** not reject $\&$not exists **then**
18:              test_combinations $\leftarrow$ candidate
19:              **for** $c\_$ in candidate **do**
20:                  taken$[c\_] = True$
21:              **end for**
22:          **end if**
23:          **if** all values of taken are *True* **then**
24:              missing_attributes $\leftarrow False$
25:              $k \leftarrow k + 1$          ▷ All attributes appeared once, next iteration
26:          **end if**
27:      **end while**
28: **end for**
29: To fill the list train_combinations, substitute in line 2 n_test_combinations to n_train_combinations. Change declaration of line 12 into line 30. Repeat from 2-28.
30: exists $=$ (candidate in test_combinations) OR (candidate in train_combinations)

---

The additional variable reject and the loop in 14-16 is to exclude a new tuple if it contains a list of attributes which have already appeared, *e.g.,* if test_combinations $=$ $[(3, \text{yellow}, \text{bright}, \text{small}), (4, \text{blue}, \text{dark}, \text{large})]$, the candidate $(3, \text{blue}, \text{dark}, \text{small})$ will be rejected. This constraint allows to guarantee that all attributes appear at least $r$ times while minimizing the number of combinations used to do so.

## A.3   **Procedure for the Generation of Training and Test Examples**

Across all the VQA-MNIST datasets, we leverage on Algorithm 1 to generate the combinations of attributes. The digits of MNIST used for the training, validation, and test splits have been fixed a

priori, with 50K training digits, 10K validation digits, and the 10K test digits of the original MNIST. Depending on the VQA task, we differentiate the data generation process.

**Attribute Extraction.**    Given a split (training or test), and the VQA question with its corresponding sub-task, we divide the objects in the split into those providing a positive and negative example. We randomly select a digit from the MNIST split which belongs to the category of the example, and we introduce the additional attributes to the image. We generate evenly examples for both cases, so to have a balanced dataset. The program layout corresponds to one of the 21 attribute instances.

**Attribute Comparison.**    The data generation repeats identically at training and testing. Given a VQA question about a relation and a split, we first collect all the pairs of objects from that split that provide a positive answer to the question. We repeat the same for the negative pairs. Then, we generate the examples based on the attributes for the pair of objects, by extracting images from the MNIST split which match with the categories of the objects. We generate an even number of positive and negative examples for each question, to have a balanced dataset.

For the tasks of attribute comparison, the program layout has a tree structure. For attribute comparison between pairs of separated objects, the sub-tasks at the leaves can be category, color, light condition, and size. These sub-tasks match with the relational questions (*eg.,* if we are asking if two objects are the same color, the leaves tackle the sub-task color). The sub-tasks and their division in groups are detailed in Table App.1. For comparison between spatial positions, the sub-tasks at the leaves can assume the form of one of the 21 attribute instances, while the root coincides with one of the four 2D spatial relations among below, above, left, and right. The sub-tasks and their division in groups are detailed in Table App.2.

# B  Implementation Details and Supplemental Results on VQA-MNIST

In Section B.1, we give the definition of image encoder and classifier architectures, common to all the NMNs trained on VQA-MNIST. We present the experimental setup and optimization details for the experiments on VQA-MNIST in Section B.2. The in-distribution generalization of the main libraries on VQA-MNIST are in Section B.3.

Then, we introduce a series of control libraries to prove the generality of our findings. In particular, in Section B.4 we compare libraries with modular image encoder to state-of-the-art libraries in the NMN literature. In Section B.5 we analyze the effect of different degrees of modularity at the image encoder stage. In Section B.6, we further verify the impact of different architectures for the intermediate modules on systematic generalization. In Section B.7, we analyze the effect of different degrees of modularity at the classifier stage. Finally, in Section B.8, we report the systematic generalization of NMNs trained ten times larger datasets than those in Figure 3.

## B.1  Description of the Image Encoder and Classifier Architectures

Table App.3: Image encoder module VQA-MNIST. The first two parameters in Conv2d refer to the number of input and output channels. For those libraries without batch normalization, the BatchNorm2d and BatchNorm1d should not be considered.

| | Image Encoder | |
|---|---|---|
| | Layer | Parameters |
| (0) | Conv2d | (3, 64, kernel_size=(3, 3), stride=(1, 1), padding=(1, 1), bias=False) |
| (1) | BatchNorm2d | (64, eps=1e-05, momentum=0.1, affine=True, track_running_stats=True) |
| (2) | ReLU | |
| (3) | Conv2d | (64, 64, kernel_size=(3, 3), stride=(1, 1), padding=(1, 1), bias=False) |
| (4) | BatchNorm2d | (64, eps=1e-05, momentum=0.1, affine=True, track_running_stats=True) |
| (5) | ReLU | |
| (6) | MaxPool2d | (kernel_size=2, stride=2, padding=0, dilation=1, ceil_mode=False) |
| (7) | Conv2d | (64, 64, kernel_size=(3, 3), stride=(1, 1), padding=(1, 1), bias=False) |
| (8) | BatchNorm2d | (64, eps=1e-05, momentum=0.1, affine=True, track_running_stats=True) |
| (9) | ReLU | |
| (10) | Conv2d | (64, 64, kernel_size=(3, 3), stride=(1, 1), padding=(1, 1), bias=False) |
| (11) | BatchNorm2d | (64, eps=1e-05, momentum=0.1, affine=True, track_running_stats=True) |
| (12) | ReLU | |
| (13) | MaxPool2d | (kernel_size=2, stride=2, padding=0, dilation=1, ceil_mode=False) |
| (14) | Conv2d | (64, 64, kernel_size=(3, 3), stride=(1, 1), padding=(1, 1), bias=False) |
| (15) | BatchNorm2d | (64, eps=1e-05, momentum=0.1, affine=True, track_running_stats=True) |
| (16) | ReLU | |
| (17) | Conv2d | (64, 64, kernel_size=(3, 3), stride=(1, 1), padding=(1, 1), bias=False) |
| (18) | BatchNorm2d | (64, eps=1e-05, momentum=0.1, affine=True, track_running_stats=True) |
| (19) | ReLU | |

The architectures of the image encoder and the classifier modules are shared across all the experiments on VQA-MNIST. All the libraries in Section 4, as well as most of those in this section, do not have batch normalization, except for *all(bn) - all - all(bn)* and *all(bn) - sub-task - all(bn)* libraries. We report the implementation of an image encoder module and a classifier module, respectively in Table App.3 and Table App.4, where we include the batch normalization.

For those image encoders and classifiers without batch normalization, the BatchNorm2d and BatchNorm1d layers need to be excluded (respectively transformations 1, 4, 8, 11, 15, and 18 in the image encoder and transformations 1 and 6 in the classifier). Notice that in Tables App.3 and App.4, the first two parameters for the Conv2d define the number of input and output channels.

## B.2  Optimization Details

Across all the experiments on VQA-MNIST on $210K$ training examples, we fix the number of training steps to $200K$. For the task of attribute extraction from a single object, the grid of hyper-

Table App.4: Classifier module for libraries VQA-MNIST. The first two parameters in Conv2d refer to the number of input and output channels. For those libraries without batch normalization, the BatchNorm2d and BatchNorm1d should not be considered.

| | Binary Classifier | |
|---|---|---|
| | Layer | Parameters |
| (0) | Conv2d | (64, 512, kernel_size=(1, 1), stride=(1, 1)) |
| (1) | BatchNorm2d | (512, eps=1e-05, momentum=0.1, affine=True, track_running_stats=True) |
| (2) | ReLU | |
| (3) | MaxPool2d | (kernel_size=7, stride=7, padding=0, dilation=1, ceil_mode=False) |
| (4) | Flatten | |
| (5) | Linear | (in_features=512, out_features=1024, bias=True) |
| (6) | BatchNorm1d | (1024, eps=1e-05, momentum=0.1, affine=True, track_running_stats=True) |
| (7) | ReLU | |
| (8) | Linear | (in_features=1024, out_features=2, bias=True) |

parameters includes batch size with candidate values $[128, 256]$, and the learning rates with values $[10^{-4}, 10^{-3}, 0.005, 0.01]$. Given the high computational cost of training, for all the remaining experiments we reduce the number of models by fixing the batch size to $64$ and the learning rate hyper-parameters to the values $[10^{-5}, 10^{-4}, 10^{-3}, 0.005, 0.01]$.

During training, the learning rate is kept constant. We notice that in some cases, especially on comparison tasks, the training loss function, as well as the training accuracy, often presents several plateaus alternated to jumps, despite the absence of a decaying learning rate. This did not allow us to use early stopping as a strategy to accelerate the experiments.

### B.3 In-distribution generalization

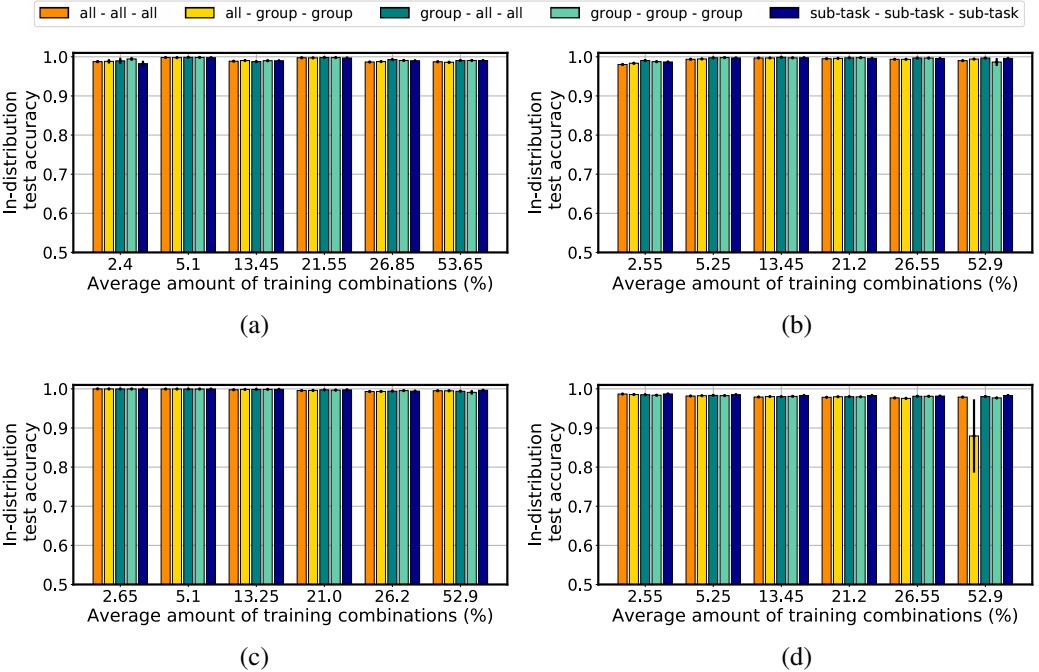

Figure App.2: In-distribution generalization on the VQA-MNIST dataset, for (a) attribute extraction from single object, (b) attribute extraction from multiple objects, (c) attribute comparison between pairs of separated objects, and (d) comparison between spatial positions.

We aim at comparing the in-distribution and systematic generalization trends for the libraries previously tested on novel combinations, in Figure 3. The plots in Figure App.2 shows the in-distribution generalization for (a) attribute extraction from single object, (b) attribute extraction from multiple objects, (c) attribute comparison between pairs of separated objects, and (d) comparison between spatial positions. The disparity between testing in-distribution generalization or systematic generalization is dramatic. In absence of a bias, the tasks of attribute extraction and comparison, with the given amount of training examples looks easy. Indeed, the plots across the four VQA tasks show that all the libraries represent equivalently a good choice to achieve generalization within in-distribution testing examples. This result emphasizes the importance of measuring systematic generalization as a strategy to highlight the limited capability of models to generalize to novel combinations.

### B.4 Results for Additional state-of-the-art Libraries and Comparison with Libraries with Modular Image Encoder

With the exception of the *all - all - all* library, the libraries in Figure 1a are not the standard ones from the literature. To confirm our findings on the superiority of the modular image encoder (*group - all - all* in particular) with previous works, we add the *all - sub-task - all* library used in [13]. Its implementation consists of a single image encoder and a single classifier, and Residual intermediate modules. Given the usage of a single module at the image encoder and the classifier stages, *all - all - all* library and *all - sub-task - all* library have variants with batch normalization at both the image encoder and classifier [11]. We denote these libraries as *all(bn) - all - all(bn)* and *all(bn) - sub-task - all(bn)*. See Figure App.3a for the depictions of these libraries of modules.

We measure the systematic generalization of those libraries. In Figure App.3b-e, we reported the performance on (b) attribute extraction from single object, (c) attribute extraction from multiple objects, (d) attribute comparison between pairs of separated objects, and (e) comparison between spatial positions. The systematic generalization accuracy for state-of-the-art methods remains clearly below the performance of the *group - all - all* library.

### B.5 Results for Libraries with Modular Image Encoder

Results in Figure 3 highlight the importance of a library of modules at the image encoder stage, together with a medium degree of modularity. To further investigate this aspect, we introduce the *sub-task - all - all* library, an additional library with image encoder modules per each sub-task. We further define two additional libraries for the task of attribute comparison, reported in Figure App.4b. There, given the tree program layout for this task, we rely on libraries with different degrees of modularity at the intermediate modules stage: at the leaves, we use a module per sub-task, while the root has an intermediate module per group of sub-task, where the division of sub-tasks follows the one in Table App.1. The main difference in the libraries of Figure App.4b is at the image encoder stage. On the left, the *all - group/sub-task - all* library relies on a single image encoder module. On the right, the *sub-task - group/sub-task - all* library relies on the largest library of image encoder modules. These two libraries have a single classifier module.

We tested the libraries in the first three columns of Figure App.4a on the VQA-MNIST tasks, while the libraries on the last two columns are used on the task of attribute comparison between a pair of separated objects. In Figure App.5a-d, we reported the systematic generalization performance for the tasks of (a) attribute extraction from single object, (b) attribute extraction from multiple objects, (c) attribute comparison between pairs of separated objects, and (d) comparison between spatial positions. We observe that the *group - all - all* library consistently outperforms the *sub-task - all - all* library across all the tasks. The *sub-task - all - all* library is nonetheless superior to the standard *all - all - all* library. In (c), the modularity of the image encoder library brings advantage also to the network with different degrees of modularity at the intermediate stage.

### B.6 Results for Libraries for Different Implementations of Intermediate Modules per sub-task

To exclude that the different implementation of the Residual from the Find module could be a reason for the poorer systematic generalization performance of the *sub-task - sub-task - sub-task* library, we introduce a similar library, which we call *sub-task - sub-task(Find) - sub-task* library. In the latter, the

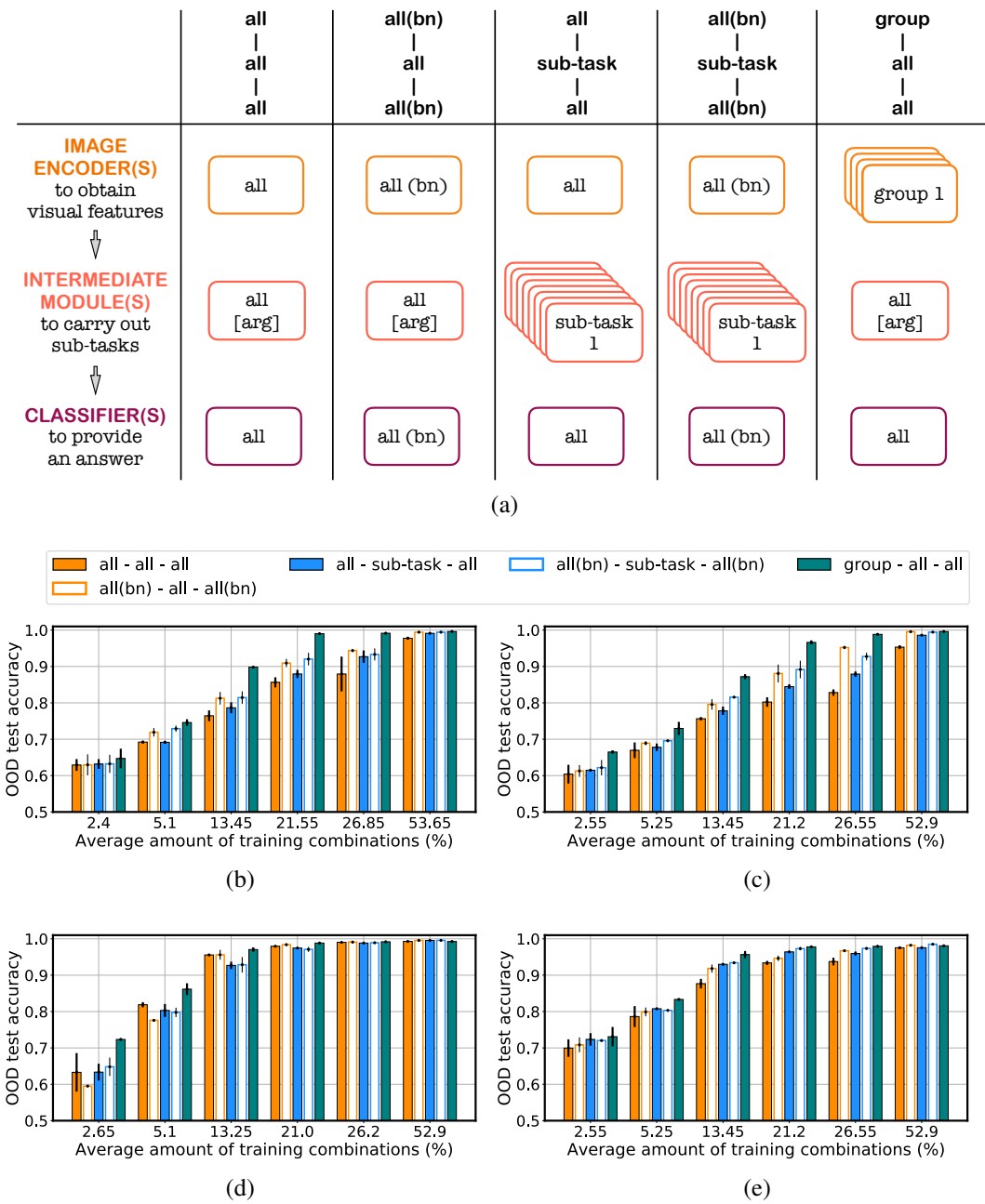

Figure App.3: Comparison of libraries with modular image encoder and state-of-the-art libraries. OOD test accuracy refers to systematic generalization accuracy. (a) State-of-the-art libraries and library with modular image encoder (*group - all - all*). Systematic generalization accuracy on (b) attribute extraction from single object, (c) attribute extraction from multiple objects, (d) attribute comparison between pairs of separated objects, and (e) comparison between spatial positions.

implementation of the intermediate modules rely on the Find module as in Equations 5-6, where each sub-task defines a group. See Figure App.6a for the depictions of these libraries of modules.

Figure App.6b-e shows the systematic generalization of the libraries of Figure App.6a, trained on (b) attribute extraction from single object, (c) attribute extraction from multiple objects, (d) attribute comparison between pairs of separated objects, and (e) comparison between spatial positions. We observe that libraries with sub-task degree of modularity at the image encoder stage perform quite

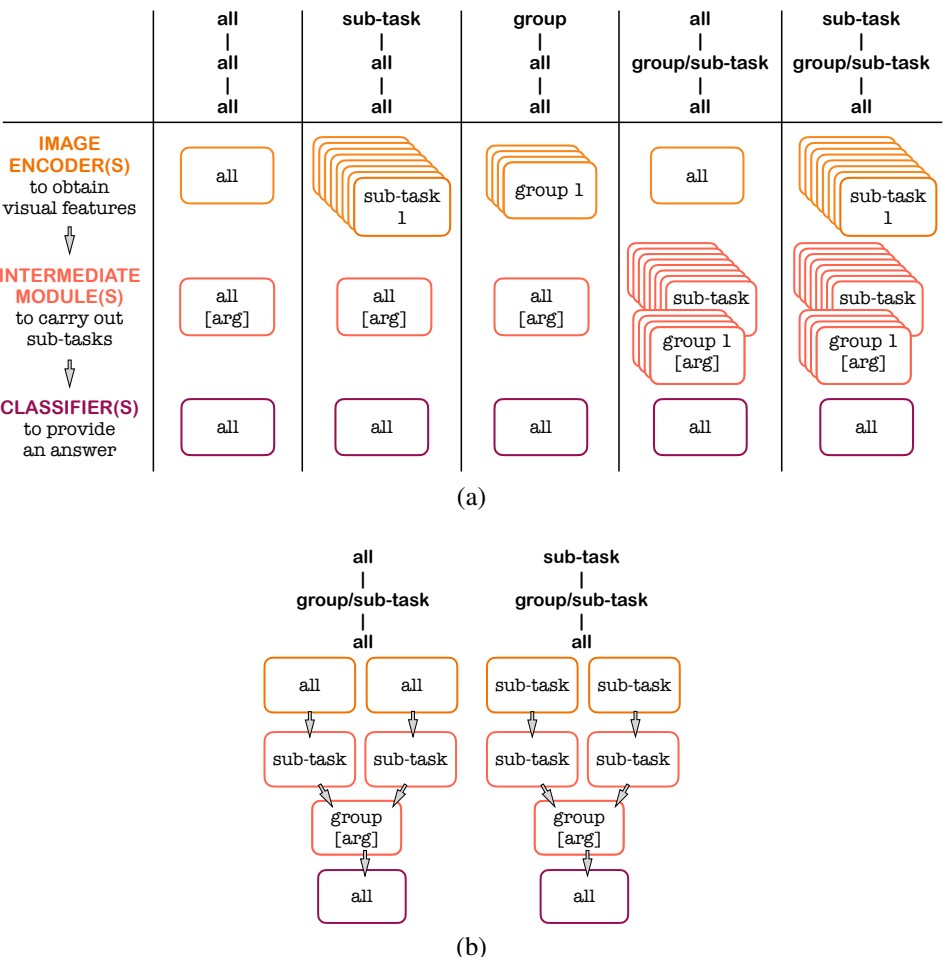

Figure App.4: Comparison between libraries with different degrees of modularity at the image encoder stage. (a) Libraries with different degrees of modularity at the image encoder stage, (b) Implementation of libraries with group/sub-task libraries at intermediate module stage.

similar independently from the implementation of their intermediate modules, and show consistently lower performance than the *group - all - all* library.

## B.7    Results for Libraries with Modular Classifier

We further consider the effect of libraries with different degrees of modularity at the classifier stage on systematic generalization. To do so, we introduce two libraries with a shared image encoder module and shared intermediate modules. As before, we envision three degrees of modularity at the classifier stage: shared module, that is the *all - all - all* library, per group of sub-tasks, that is, the *all - all - group* library, and per sub-task, that is, the *all - all - sub-task* library. Figure App.7a depicts these libraries.

The systematic generalization performance for the libraries in Figure App.7a across the VQA-MNIST tasks is reported in Figure App.7b-e. The conclusion here is that the degree of modularity of the library at the classifier stage alone does not bring any advantages across the tested tasks.

## B.8    Results on More Training Examples

We performed additional experiments to check if the amount of training examples plays a role in the trade-off among degrees of modularity. With this goal, we increased by a factor ten the amount of

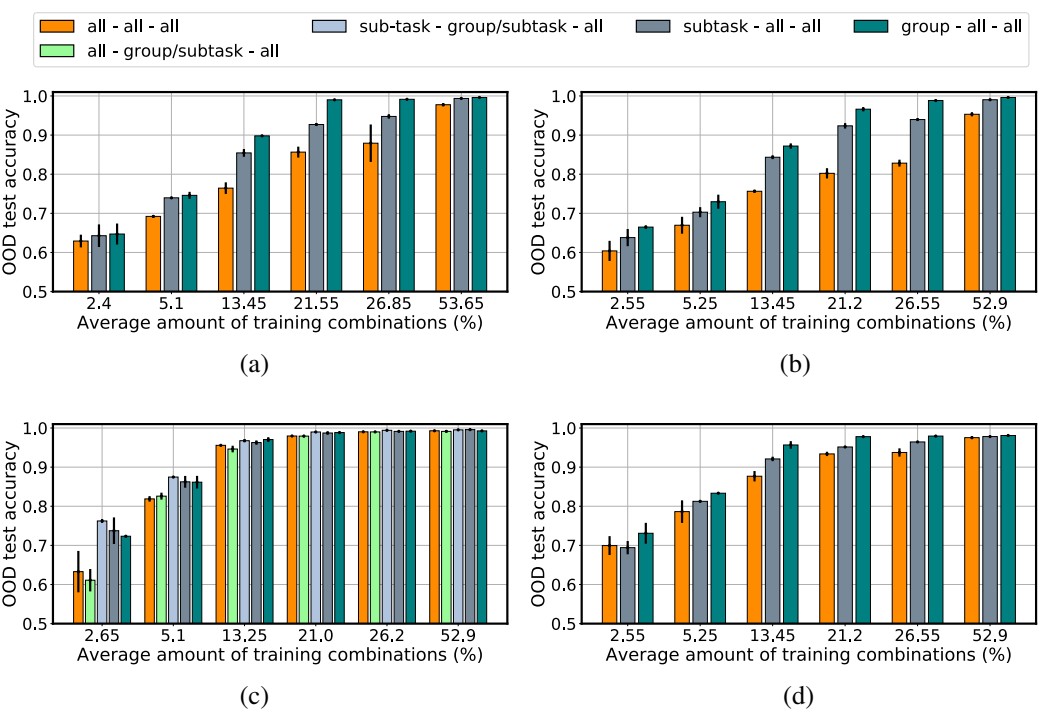

Figure App.5: Results for comparison between libraries with different degree of modularity at the image encoder stage. OOD test accuracy refers to systematic generalization accuracy. (a-d) Systematic generalization accuracy on (a) attribute extraction from single object, (b) attribute extraction from multiple objects, (c) attribute comparison between pairs of separated objects, and (d) comparison between spatial positions.

training examples, in Figure App.8, for (a) attribute extraction from single object, and (b) attribute comparison between pairs of separated objects.

The NMNs trained on (a) have a fixed batch size to value 64 and a grid of learning rates with values $[10^{-5}, 10^{-4}, 0.001, 0.005, 0.01]$. We fixed at half a million the amount of iterations. The NMNs trained on dataset (b) share the same set of hyper-parameters of the case with ten times less data. We fixed the number of iterations to one million, but due to the high computational cost, we ran the experiments for maximum $\sim 300$ hours on a NVIDIA DGX-1 machine. Not all the models reached the last iteration, but all their loss curves qualitatively show convergence. We report the results in Figure App.8. (a) shows the case of attribute extraction from a single objects, while (b) reports the performance for attribute comparison between pairs of objects.

The colored bars report the systematic generalization performance over one trial, with 2.1 million training examples, while the black dots display the systematic generalization performance for the corresponding model trained on ten times less examples. We observe that, consistently with the previous results, the *group - group - group* library outperforms the other libraries, in Figure App.8a. In Figure App.8b, the NMN with modular library at the image encoder stage achieve the highest systematic generalization. This result shows that high systematic generalization is not reachable by increasing the amount of training data. This conclusion is consistent with the observation in previous works on out-of-distribution generalization [9, 14].

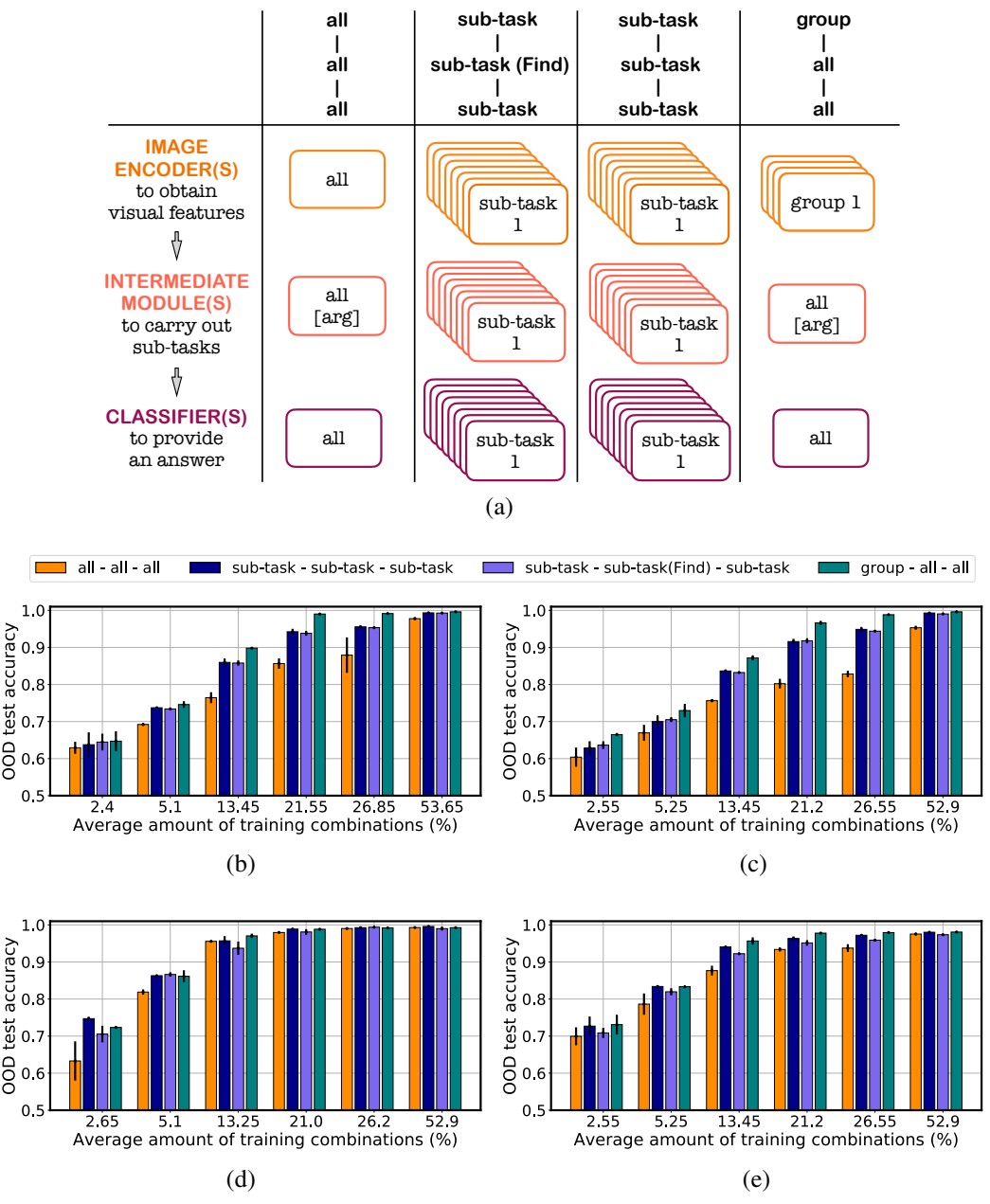

Figure App.6: Comparison between libraries of same degree of modularity, but different implementation of the intermediate modules. OOD test accuracy refers to systematic generalization accuracy. (a) Two libraries with different implementation of the intermediate modules and *all - all - all* and *group - all - all* libraries for reference. Systematic generalization accuracy on (b) attribute extraction from single object, (c) attribute extraction from multiple objects, (d) attribute comparison between pairs of separated objects, and (e) comparison between spatial positions.

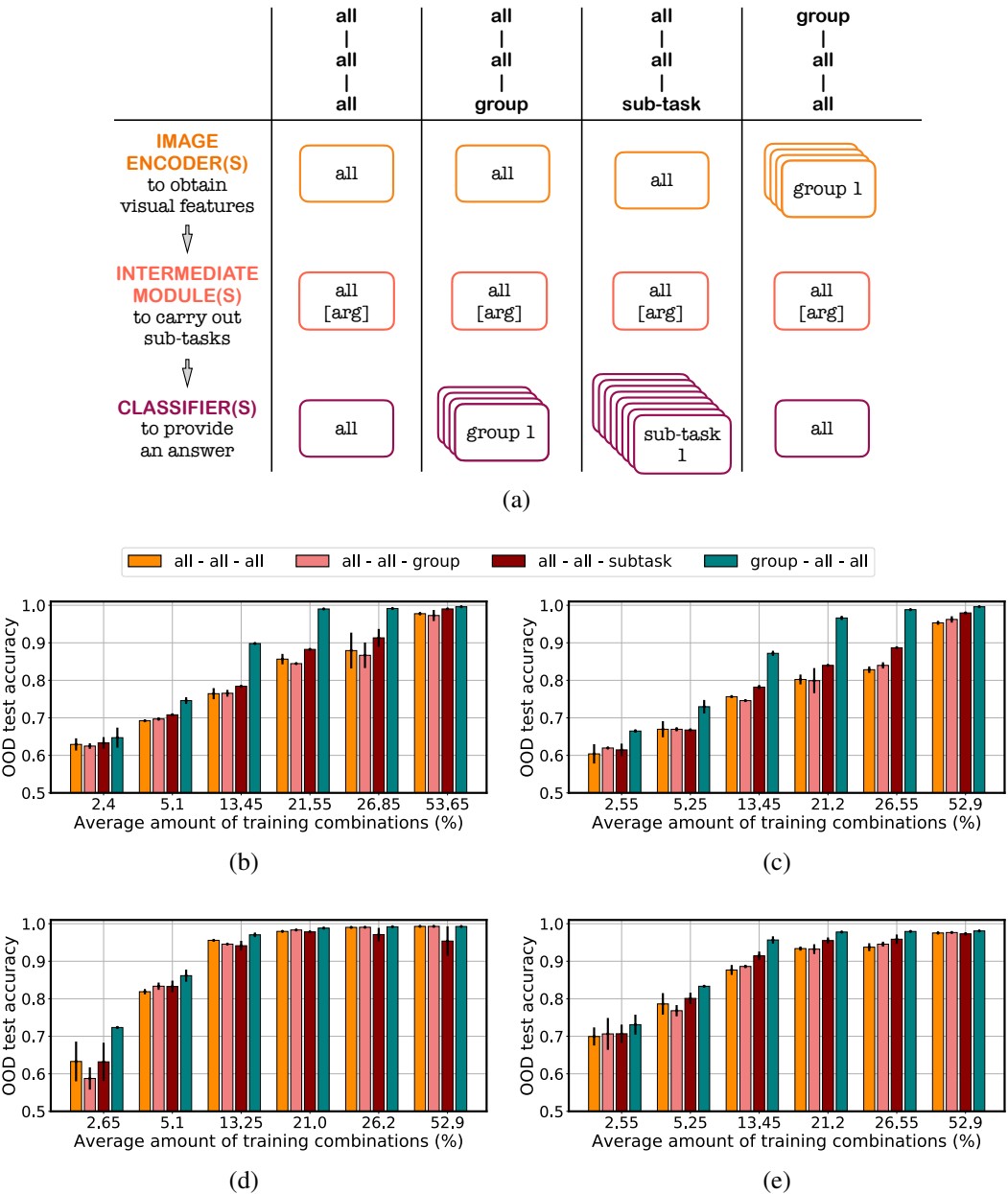

Figure App.7: Comparison between libraries with different degrees of modularity at the classifier stage. OOD test accuracy refers to systematic generalization accuracy. (a) Three libraries with different degrees of modularity at the classifier stage and the *group - all - all* library for reference. Systematic generalization accuracy on (b) attribute extraction from single object, (c) attribute extraction from multiple objects, (d) attribute comparison between pairs of separated objects, and (e) comparison between spatial positions.

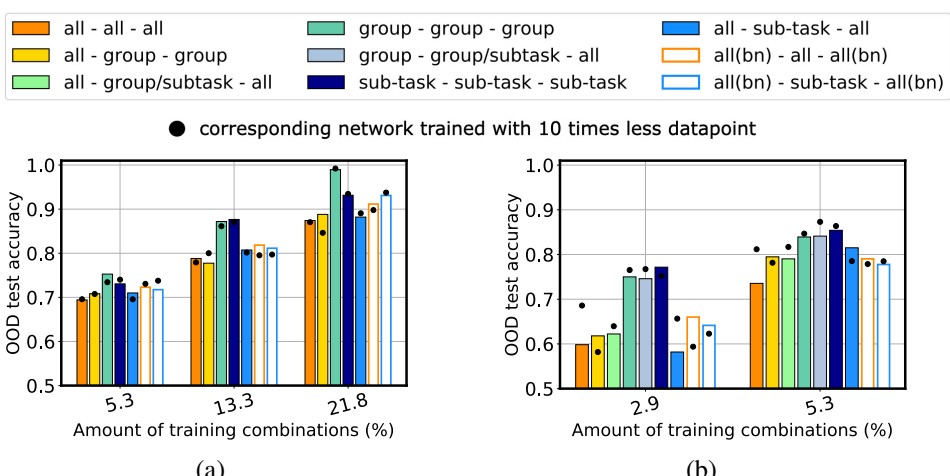

Figure App.8: Experiments with ten times more data. OOD test accuracy refers to systematic generalization accuracy. The colored bars correspond to networks trained with 2.1 million images, dots correspond to the case of $210K$ images. Systematic generalization accuracy for (a) attribute extraction from single object, (b) attribute comparison between pairs of separated objects.

# C   Supplemental Results on SQOOP

To perform these experiments, we rely on the networks proposed in [11]. Both tree-NMNs make use of batch normalization in their image encoder and classifier stages.

To validate that NMNs outperform other VQA models in systematic generalization, we report in Table App.5 the results obtained on SQOOP for two non-modular neural networks, namely re-implementation of MAC [20] from Bahdanau *et al.* [11] and FiLM [19], with the results of tree-NMNs we have reported in Table 1. We adopted the same hyper-parameters from [11].

For each model, we observe a wide gap in systematic generalization on the two datasets. For the FiLM architecture, reducing the amount of objects in the image leads to higher systematic generalization than for the case of five objects, while the opposite happens to MAC.

A direct comparison between non-modular networks and NMN is not possible, given that the NMN are provided with an optimal program layout. Nonetheless, the higher systematic generalization of NMNs highlights the potential benefit in the use of NMNs wherever there is a mechanism to identify a proper program, given a question.

Table App.5: Systematic generalization accuracy (%) for MAC (reimplemented by Bahdanau *et al.* [11]), FiLM, and NMN architectures with tree layouts. Top row: bias in the question (five objects per image). Bottom row: bias in the co-occurrence of objects in the image (two objects per image).

| Bias type | MAC (Bahdanau *et al.* [11]) | FiLM | *all - all - all* with bn | *all - sub-task - all* with bn |
|---|---|---|---|---|
| Five objects per image (as in [11]) | $84 \pm 4$ | $70 \pm 2$ | $\mathbf{99.8 \pm 0.2}$ | $\mathbf{99.96 \pm 0.06}$ |
| Two objects per image | $66 \pm 5$ | $77 \pm 3$ | $84 \pm 2$ | $\mathbf{88.5 \pm 0.5}$ |

# D Implementation Details and Supplemental Experiments on CLEVR-CoGenT

In Appendix D.2 we report the implementation of the Vector-NMN as in [5], with its image encoder, intermediate module, and classifier. Then, in Appendix D.1 we describe the Compositional Generalization Test (CoGenT), its tasks, and our grouping of sub-tasks, in Appendix D.3. Lastly, we report the results obtained on single questions for other non-modular networks, and different NMN implementations.

## D.1 CLEVR-CoGenT

The Compositional Generalization Test (CLEVR-CoGenT) dataset [2] includes training, validation, and test splits. The data generation process follows the usual criteria of CLEVR, but CoGenT is built to explicitly measure the ability of a model to generalize to novel combinations of objects and attributes. CLEVR-CoGenT has two conditions: Condition A and Condition B. In Condition A, the one used at training, cubes, cylinders and spheres appear in different sizes, colors, and positions. Cubes are gray, blue, brown, or yellow, cylinders are red, green, purple, or cyan, while spheres can have any color. In Condition B these combinations of attributes are reversed. Cubes are red, green, purple, or cyan, cylinders are gray, blue, brown, or yellow, and spheres can have any color. Condition B is the split to measure systematic generalization of a model. The models can be validated and tested in-distribution as well.

## D.2 Vector-NMN

We define Vector-NMN as in previous work [5]. On the family of CLEVR datasets, the image encoder of Vector-NMN takes in input the output features from a ResNet-101 as detailed in Section 4.1 of [2]. The image encoder consists of two convolutional layers with ReLU activation, as detailed in Table App.6, the classifier is a multi-layer perceptron, as detailed in Table App.7.

Table App.6: Image encoder of Vector-NMN.

| | Image encoder | |
|---|---|---|
| (0) | Conv2d | ((1024, 64, kernel_size=(3, 3), stride=(1, 1), padding=(1, 1)) |
| (1) | ReLU | |
| (2) | Conv2d | (64, 128, kernel_size=(3, 3), stride=(1, 1), padding=(1, 1)) |
| (3) | ReLU | |

Table App.7: Classifier of Vector-NMN.

| | Classifier | |
|---|---|---|
| (0) | Flatten | |
| (1) | Linear | (in_features=128, out_features=1024, bias=True) |
| (2) | ReLU | |
| (3) | Linear | (in_features=1024, out_features=32, bias=True) |

The Vector-NMN leverages on modulation through a set of weights, similarly to the FiLM network [19]. Every intermediate module of the Vector-NMN receives in input the concatenation of the embedding for the sub-task and the output from previous modules, defined as $h_c = [\text{Emb}(k); s_x; s_y]$. We define the output of the image encoder given the input features of an image as $s_{\text{img}}$. In the following, we report the definition of Vector-NMN [5]:

$$[\beta_j, \tilde{\gamma}_j] = W_2^j(\text{ReLU}(W_1^j h_c + b_1^j) + b_2^j), \quad \text{with } j \in [0, 1], \tag{App.1}$$

$$\gamma_j = 2\tanh(\tilde{\gamma}_j) + 1, \tag{App.2}$$

$$\tilde{h}_1 = \text{ReLU}(U_1 * (\gamma_1 \odot s_{\text{img}} \oplus \beta_1)), \tag{App.3}$$

$$\tilde{h}_2 = \text{ReLU}(U_2 * (\gamma_2 \odot \tilde{h}_1 \oplus \beta_2) + h_x), \tag{App.4}$$

$$h_i(k, s_x, s_y) = \text{maxpool}(\tilde{h}_2). \tag{App.5}$$

### D.3 Grouping of sub-tasks

We decide to divide sub-tasks based on their similarity. We group together all the sub-tasks which require determining colors, shapes, materials, etc. We identify eight groups of sub-tasks. In the following, we specify each group, each with its related sub-tasks: counting, with `count`, `equal_integer`, `exist`, `greater_than`, `less_than`, `unique`; colors, with `equal_color`, `filter_color[*color instance]`, `query_color`, `same_color`; materials, with `equal_material`, `same_material`, `filter_material[*material instance]`, `query_material`; shapes, with `equal_shape`, `filter_shape[*shape instance]`, `query_shape`, `same_shape`; sizes, with `equal_size`, `filter_size[*size instance]`, `query_size`, `same_size`; spatial relations `relate[* four possible directions]`; logical operations `intersection`, `union`; and a group neutral to sub-tasks, with `scene`.

### D.4 Supplemental Results on CLEVR-CoGenT

We now report the systematic generalization performance on CLEVR-CoGenT. The results include non-modular networks, as MAC [20] re-implemented by Bahdanau *et al.* [5], its variant with batch normalization in the image encoder and classifier, the FiLM network [19], the Vector-NMN and the Tensor-NMN, their variant with batch normalization in the image encoder and classifier, and an additional NMN which receives the input $s_{img}$ at every intermediate step.

We divide the questions into several question types, by considering the first sub-task of the program layout for each question, as in [2]. In Table App.8, we report separately in a scale from 0 to 100 (*ie.* percentage) the performance for each model and each question type. It is important to underline that all the NMNs are trained and tested using the ground-truth program layout.

The best network for each question type is highlighted in bold. On nine over thirteen question types, our Vector-NMN with modular image encoder has higher performance than pre-existing modular networks, given the same ground-truth program layout. Modularity of the image encoder is particularly convenient on the hardest type of questions, as `query_shape` and `query_color`.

Table App.8: Systematic generalization accuracy (%) on validation split of CLEVR-CoGenT Condition B across different question types.

| | MAC with bn (Bahdanau *et al.* [5]) | MAC (Bahdanau *et al.* [5]) | FiLM |
|---|---|---|---|
| count | $69.5 \pm 0.5$ | $68.8 \pm 0.9$ | $\mathbf{73.3 \pm 0.9}$ |
| equal_color | $78.5 \pm 0.8$ | $79.1 \pm 0.7$ | $78.1 \pm 0.7$ |
| equal_integer | $81 \pm 1$ | $79 \pm 2$ | $82 \pm 2$ |
| equal_material | $77 \pm 3$ | $80.6 \pm 0.8$ | $82 \pm 2$ |
| equal_shape | $96 \pm 1$ | $92 \pm 2$ | $\mathbf{97.0 \pm 0.7}$ |
| equal_size | $77 \pm 2$ | $81.3 \pm 0.7$ | $81 \pm 1$ |
| exist | $84.9 \pm 0.4$ | $85.1 \pm 0.6$ | $\mathbf{86.8 \pm 0.4}$ |
| greater_than | $83.4 \pm 0.9$ | $83 \pm 1$ | $83.8 \pm 0.5$ |
| less_than | $81.4 \pm 0.4$ | $81.4 \pm 0.7$ | $81.8 \pm 0.8$ |
| query_color | $63 \pm 1$ | $\mathbf{68 \pm 1}$ | $65.5 \pm 0.3$ |
| query_material | $85 \pm 1$ | $87 \pm 1$ | $87 \pm 1$ |
| query_shape | $34.7 \pm 0.7$ | $36 \pm 1$ | $36.2 \pm 0.6$ |
| query_size | $84 \pm 2$ | $86 \pm 1$ | $85.4 \pm 0.5$ |

| | Tensor simple arch | Tensor | Tensor with bn |
|---|---|---|---|
| count | $69.7 \pm 0.5$ | $69.7 \pm 0.8$ | $70.9 \pm 0.7$ |
| equal_color | $76 \pm 2$ | $75.6 \pm 0.8$ | $78.0 \pm 0.7$ |
| equal_integer | $83 \pm 1$ | $82.7 \pm 0.3$ | $83.3 \pm 0.7$ |
| equal_material | $74 \pm 2$ | $74 \pm 2$ | $75 \pm 1$ |
| equal_shape | $92 \pm 2$ | $91 \pm 2$ | $92 \pm 3$ |
| equal_size | $76 \pm 3$ | $75 \pm 1$ | $77 \pm 1$ |
| exist | $83.8 \pm 0.3$ | $84.2 \pm 0.4$ | $84.9 \pm 0.2$ |
| greater_than | $84.4 \pm 0.5$ | $83.8 \pm 0.6$ | $84.4 \pm 0.5$ |
| less_than | $82.2 \pm 0.3$ | $80.7 \pm 0.9$ | $82 \pm 1$ |
| query_color | $59 \pm 1$ | $58 \pm 1$ | $62.6 \pm 0.9$ |
| query_material | $85 \pm 1$ | $84.1 \pm 0.9$ | $85.7 \pm 0.7$ |
| query_shape | $37 \pm 2$ | $37 \pm 1$ | $37 \pm 1$ |
| query_size | $85 \pm 1$ | $83.5 \pm 0.6$ | $84.9 \pm 0.8$ |

| | Vector | Vector with bn | Vector modular (**ours**) |
|---|---|---|---|
| count | $70.4 \pm 0.4$ | $71.5 \pm 0.9$ | $71 \pm 1$ |
| equal_color | $74 \pm 1$ | $76 \pm 2$ | $\mathbf{80 \pm 1}$ |
| equal_integer | $78 \pm 2$ | $81.6 \pm 0.9$ | $\mathbf{85 \pm 2}$ |
| equal_material | $74.2 \pm 0.7$ | $77 \pm 3$ | $\mathbf{84 \pm 2}$ |
| equal_shape | $89 \pm 3$ | $86 \pm 2$ | $79 \pm 2$ |
| equal_size | $75 \pm 1$ | $78 \pm 3$ | $\mathbf{88 \pm 2}$ |
| exist | $84.4 \pm 0.4$ | $85 \pm 0.7$ | $84.4 \pm 0.5$ |
| greater_than | $83.6 \pm 0.4$ | $84 \pm 1$ | $\mathbf{89 \pm 1}$ |
| less_than | $82.0 \pm 0.5$ | $83 \pm 1$ | $\mathbf{87 \pm 2}$ |
| query_color | $60 \pm 1$ | $66 \pm 5$ | $67 \pm 4$ |
| query_material | $84.7 \pm 0.4$ | $86 \pm 2$ | $\mathbf{88.2 \pm 0.8}$ |
| query_shape | $40 \pm 3$ | $41 \pm 2$ | $\mathbf{52 \pm 3}$ |
| query_size | $84.7 \pm 0.7$ | $86 \pm 2$ | $\mathbf{89.5 \pm 0.5}$ |

# E    Supplemental Results on CLOSURE

CLOSURE is a novel test set for models trained on the CLEVR dataset [5]. This dataset introduces seven question templates that have high overlap to the CLEVR questions but zero probability of appearing under the CLEVR data distribution.

We trained Tensor-NMN, Vector-NMN and our Vector-NMN with modular image encoder on the CLEVR dataset by keeping the hyper-parameters (learning rate, batch size, number of iterations, network's number of layers and transformations) as in [5]. For our Vector-NMN, the separation of sub-tasks into groups is specified in Section 5.

The results in Table App.9 shows the performance of our Vector-NMN with modular image encoder compared to the original Vector-NNM and Tensor-NMN, for novel questions templates of CLOSURE.

Table App.9: Systematic generalization accuracy (%) on each question type of CLOSURE for Tensor-NMN and Vector-NMN (as reported in [5]), and for our Vector-NMN, across five repetitions.

|  | Tensor-NMN | Vector-NMN | Vector-NMN with modular image encoder (ours) |
|---|---|---|---|
| and_mat_spa | $64.9 \pm 2$ | $\mathbf{86.3 \pm 2.5}$ | $65.8 \pm 2.4$ |
| or_mat | $44.8 \pm 6.8$ | $\mathbf{91.5 \pm 0.77}$ | $70.3 \pm 3.3$ |
| or_mat_spa | $47.9 \pm 5.8$ | $\mathbf{88.6 \pm 1.2}$ | $65.2 \pm 4.5$ |
| embed_spa_mat | $98.1 \pm 0.38$ | $\mathbf{98.5 \pm 0.13}$ | $88.6 \pm 1.8$ |
| embed_mat_spa | $79.3 \pm 0.83$ | $\mathbf{98.7 \pm 0.19}$ | $94.0 \pm 1.4$ |
| compare_mat | $90.7 \pm 1.8$ | $\mathbf{98.5 \pm 0.17}$ | $89.8 \pm 1.7$ |
| compare_mat_spa | $91.2 \pm 1.9$ | $\mathbf{98.4 \pm 0.3}$ | $90.2 \pm 2.1$ |

The Vector-NMN outperforms our modular Vector-NMN across all questions. These two models only differ at the image encoder stage, which seem to suggest that a limited distribution of questions can have a different effect on modularity from what we observed in previous experiments. Note that this experiment is the only experiment in the paper in which modularity at the image encoder stage does not improve systematic generalization. This experiment is also the only one in which the distribution of program layouts in the training and testing is different. This is an interesting phenomenon which requires of further analysis (it is unclear how general this phenomenon is, what are the trends for different amounts of bias, etc.). A possible explanation is that having more modules leads to a more diverse set of possible networks, and if there is bias in the program layout, the difference between the trained networks and tested ones may become larger. This suggests that there is a trade-off between modularity and bias in the program layout.

# F  Code and Computational Cost

All the code for data generation, data loading, training, testing, and the networks trained on CLEVR-CoGenT can be found at this link: `https://github.com/vanessadamario/understanding_reasoning.git`. This work stems from two forked repositories which investigated systematic generalization in VQA: `https://github.com/rizar/systematic-generalization-sqoop.git` [11] and `https://github.com/rizar/CLOSURE.git` [5]. Our code, as the one from which we took inspiration, is open access and reusable.

The experiments have been run on multiple platforms (AWS + MIT's OpenMind Computing Cluster[1]).

**VQA-MNIST**  Across all the experiments with 210K training examples, the number of epochs is always higher than 60 (more specifically, $(200000*64)/210000 > 60$). Across all the experiments, we observed that the training converged for such number of epochs.

The resources and amount of time for VQA-MNIST experiments varies: with larger batch size, the experiments of attribute extraction from single object (experiment_1) took around ten days to complete the training of each data-run (NVIDIA Tesla K80, NVIDIA Tesla K40, NVIDIA TitanX).

We moved the experiments for attribute comparison and attribute extraction from multiple objects to AWS. We further parallelized those using Ray (`https://ray.io/`), a parallelization framework. On AWS, we have been using 60 instances of g4dn.2xlarge, with 32 GB RAM, Intel Custom Cascade Lake CPU, NVIDIA T4 GPU. Training a data-run for these experiments took less than 48 h.

For the experiments with 2.1M training examples, we use an NVIDIA DGX-1. For the attribute extraction from a single object (500K iterations, batch size 64), training all models (for grid search) in parallel (through Ray we were training ten NMNs at the time) took about two weeks of computation. For the attribute comparison between separated objects ($1M$ iterations, batch size 64), training all models (for grid search) in parallel (ten experiments at the time), was taking more than two weeks. Experiments that took longer than that time were interrupted, but all the training curves reached a plateau at that point.

**SQOOP**  The experiments on SQOOP datasets took less than 15 hours on a single GPU (NVIDIA Tesla K80, NVIDIA Titan X).

**CLEVR datasets**  The experiments on the CLEVR dataset have an average training time of a week, which varies depending on the network. These models are trained on single GPU (NVIDIA Tesla K80, NVIDIA Tesla K40, NVIDIA Titan X).

---

[1] `https://openmind.mit.edu/`