# OpenReview forum: "How Modular should Neural Module Networks Be for Systematic Generalization?"
_NeurIPS.cc/2021/Conference — NeurIPS 2021 Poster_

### Official Review · Reviewer_1KRM · 2021-07-06

**Rating:** 7
**Confidence:** 4

**Summary:**

The authors analyze the preferred granularity of the modules at different stages of modular neural networks, which leads to the best systematic generalization. Based on Multi-attribute MNIST, they conclude that the modules are most useful when responsible to a specific semantically meaningful feature (like color, shape, etc), as opposed to having a single module or having modules for specific instances of those features (like green, 2, etc). They also conclude, somewhat surprisingly, that modularity is important already in the early stages of the network, so the feature extractor should also be modular. They show the advantage of the modular feature extractor on the more complex CLEVR dataset as well.

**Limitations And Societal Impact:**

Yes

**Main Review:**

# Originality
To the best of my knowledge, the provided analysis on the desired granularity of modules is original.

# Quality
The paper is sound and the claims are supported by the findings.

# Clarity
The paper is well written and mostly clear. A few recommendations:
- It would worth explicitly mentioning what does the subtask mean early in the paper
- For the multi-digit MNIST experiments, Figure 2b shows 2 feature extractor columns. How the multi-digit image is split among those 2 columns? Is it fed to both of them or are the 2 images rendered separately and fed to the encoders individually?

# Significance
The findings on the importance of modular encoding is surprising and the community would benefit from such a conclusion. The paper also confirms that the intuitive way of allocating different modules to different semantically meaningful attributes is beneficial. The paper thus is valuable for the community.

# Additional questions
An additional experiment that I would like to see is what is the benefit of a modular encoder alone with the rest of the network being monolithic.


**Time Spent Reviewing:**

4

---

### Official Review · Reviewer_w7Fj · 2021-07-14

**Rating:** 7
**Confidence:** 3

**Summary:**

This work investigates how the degree and location of modularity in neural module networks (NMN) impacts out-of-domain generalization on visual question answering (VQA) tasks. The authors tested NMNs on various VQA datasets (multi-attribute MNIST, SQOOP, CLEVR). On the first two, they separated the stages of the NMN so they were either shared among all tasks, shared only among groups of sub-tasks (like colours or categories), or shared only for particular subtasks (like individual colours or numbers). Group-level modularity, particularly at the encoder layer, improved out-of-domain test-set performance. Given these results, they showed that using a modular encoder also improved systematic generalization on the CLEVR dataset over a shared-encoder baseline.

**Main Review:**

Major suggestions:

- Re: Fig 2, there might be other interesting combinations of modular structure, for example, no model looked at combining all levels of modularity together in one model (something like sub-task - group - all). There would be 27 different arrangements in total (3 types of modularity for encoder, intermediate, and classifier). If these were explored, that should be mentioned. Otherwise, were there other reasons they weren’t investigated? Is there a technical difficulty I’m missing that makes some structures more difficult than others? Would be good to mention this.

- Regarding the results that modularity at the encoder level is important, I would have liked some suggestions on why this is the case? Is it something specific to the structure of these tasks? One could imagine that other tasks might require a shared encoder. Could be nice to interface with multi-tasking literature here (ie https://arxiv.org/abs/1706.05098)

- Finally, I wonder about the number of parameters in each of the models. For example, would the vector-NMN with 8 groups have 8 times the number of parameters in the ‘encoding’ stage? If so, it might be good to control for this, though I personally doubt that is what is responsible for the effect.

Minor suggestions:

- Acronyms should be defined right away (ie VQA in abstract)

- Figure 2: it is not obvious at a glance that columns in a) and b) refer to corresponding models since columns are not well aligned, should make that obvious

- The description of what neural module networks are was a bit unclear (needed to seek out other literature to get a better understanding)

- Should name the module mentioned on line 94 (maybe the group module)

- Some general structural issues, some further editing of sentence structure/grammar would help, and it was confusing that there was a Datasets section, a Results section, and then a 5th section with a new dataset/results, and table 3 title could be more clear.

- there is a tendency to treat systematic generalization in a binary fashion (networks are tested in-domain or out-of-domain). Really, this should be a graded measure, and would be good to be explicit for each dataset about exactly what features are out of domain and by how much.

Overall:

This work deals with an important topic that is relevant in many areas, namely, how does modularity interact with systematicity? The work provides some compelling evidence that modularity is important, especially at the encoding layer. Despite some of the issues outlined above I think the community would be interested in this result.



**Time Spent Reviewing:**

4

---

### Official Review · Reviewer_No9k · 2021-07-15

**Rating:** 4
**Confidence:** 3

**Summary:**

This paper studies neural module networks (NMNs) for systematic generalization on VQA tasks. Of particular interest is understanding how the degree of modularity across various stages of information processing affects the ability to generalize systematically.

To that extent three stages are distinguished: (1) the encoding stage from input image to features, (2) the intermediate stage where modules compare and relate image content according to the correct program layout, (3) and the classifier stage, which processes the final representation to answer the question. In each stage, the responsible networks can be shared across sub-tasks or they can be fully modularized by associating separate networks (having unique weights) for each sub-task. In the former case, this requires conditioning the shared network during the intermediate stage via a sub-task embedding to achieve specific behavior, i.e. as done in prior work (eg. the "find module" in [16]). Alternatively, an intermediate degree of modularity can be achieved by organizing sub-tasks according to groups (such as focusing on colors or categories) with sharing weights only among the modules within a particular group and using conditioning within a group to further specialize their functionality.

To evaluate how these different choices affect systematic generalization two families of datasets are proposed: Multi-Attribute MNIST contains attribute extraction tasks (for one or more colored digits) and comparison tasks (between objects and spatial locations). Here, the learner is provided only a few combinations of visual attributes during training and is expected to generalize systematically to all possible combinations. The second family of datasets is based on SQOOP [11] that  involves comparing visual objects spatially and provides only few combinations of objects during training. Here also a modified version is considered where the amount of combinations of objects in the scene is also limited. Towards the end two variants of CLEVR are considered.

On Multi-attribute MNIST it can be observed how additional modularity improves systematic generalization (i.e. using a fully modular setting, or an intermediate degree), and how modularity in the encoding stage (1) is particularly helpful. It is also shown how tree-NMNs [11] can be improved on SQOOP by increasing modularity at the intermediate stage, and how vector-NMNs [5] can be improved on CoGenT by inducing modularity at the encoding stage, although the opposite effect is observed on COGENT.

**Limitations And Societal Impact:**

No societal impact statement is provided. Limitations are adequately discussed.

**Main Review:**

The aim of this paper is to answer the question of how modular neural module networks should be for systematic generalization on VQA datasets. Three different kinds of modularity are distinguished ("all", "group" and "sub-task") applied to three different stages of information processing ("encoding", "intermediate", "classification").

It should be noted that the "all" configuration, using a shared network and conditioning, during the intermediate stage already makes use of the ground-truth program layout to structure information processing and thus is already quite modular (compared to say a system that directly maps images to answers). From this perspective, assessing the right level of modularity at the intermediate stage is really just a matter of balancing the amount of data that is made available to the module via weight sharing across sub-tasks with the complexity of learning a module that is adept at performing each of those sub-tasks. In the "all" case, all data is made available, but many different sub-tasks have to be learned. In the "sub-task" case only a subset of the data is made available (i.e. the '2' detector only sees data when the question contains a '2'), yet only a specific function has to be learned. Thus, it is not surprising that using "group" split across the ground-truth attribute types can provide a sweet spot. However, it is also clear that such a grouping (as well as the "sub-task" configuration) is not easily obtained in the general case. Further, the performance of NMNs when the groups are not "optimal" and do not reflect the ground-truth separation of attributes is not explored.

More interesting, thus, is understanding the benefits of modularity at the encoding level and classification level. From Figure 4 (comparing all-group-group and group-group-group) it can be seen that a grouped visual encoder tends to help, although other variations that could better confirm this are not explored. For example, it seems natural to me to compare all-all-all to group-all-all and to sub-task-all-all to really understand how modularity at the encoding level affects performance. The same holds true for the classification stage where, since the classification task is binary, it is not clear to me whether additional modularity will help. For example, can't the classifier just naively read out the final embedding produced by the intermediate stage to trivially answer the question? In any case, a comparison between all-all-all, all-all-group, and all-all-sub-task is not provided, which makes it difficult to say something about this. More generally, from Figure 4 there appears to be a trend towards more modular networks performing better, which is intuitive but also somewhat expected and therefore of limited significance in my view. The reason that this can be expected is because additional modularity at the properly level such as in "sub-task" will ensure that networks do not specialize to a particular input combination, but learn to act as stand-alone feature/relation detectors that can be composed (as is done in providing the correct program layout) to generalize systematically to novel combinations. Perhaps a surprising insight that was only briefly highlighted is that the performance of fully modular networks is still quite poor when only few combinations are provided (even when the amount of training data is increased), which I did not expect due to the amount of supervision is that provided for a configuration like sub-task-sub-task-sub-task. I was also surprised to see the overall better performance of NMNs on the relational questions compared to the attribute questions on Multi-Attribute MNIST, since I would have expected the relational questions to be more difficult.

There are some additional results provided on other datasets (SQOOP, CoGenT, and CLOSURE), but a systematic evaluation is lacking and there are additional differences that may act as confounders, which makes it difficult to draw broad conclusions. For example on SQOOP it is shown how using "sub-task" modularization at the intermediate stage is beneficial when combined with the tree-NMN method from [11], but "sub-task" modularity at the encoding stage or at the classification stage is not explored. Similarly, on CoGenT and CLOSURE the vector-NMN method from [5] is considered as well as a variation with a "group" modular image encoder by organizing sub-tasks according to groups. However, other kinds of modularity and at different stages, as was explored for Multi-attribute MNIST, are not considered. Here another difference is that the encoder does not act on the raw input image, but rather on features extracted from a pre-trained ResNet-101, which yields a different kind of modularity for the encoder compared to what was done previously. Nonetheless it can be observed that a modular image encoder is beneficial on this dataset. In contrast, in Table 9 in the appendix it can be seen how this same modification yields consistently worse performance on CLOSURE. There it is argued that this is due to the differences in program layouts during training and testing and future research is needed to understand this behavior in this case.

In general, I don't find that enough significant conclusions can be drawn from the current experiments presented in this paper and therefore argue for rejection. There is certainly potential, and a more systematic exploration of different degrees of modularity at different stages will help address this. It is also important that such an analysis covers multiple different data sets and uses the same set of methods, but without introducing other potential confounders, to allow for a good comparison. Finally, I encourage the authors to broaden the analysis to settings where "perfect" modularity is misspecified (such as by having groups that do not nicely partition along attributes), which will help understand whether these conclusions remain the same in more practical settings where that amount of supervision is not available.

**Minor comments**

* Regarding the data generation process for Multi-attribute MNIST it was not quite clear to me how the data is obtained. Algorithm 1 in the appendix provides pseudo-code for sampling train and test-combinations where it is ensured that test-combinations do not appear in the training combination set. However, it is not clear to me how the fraction of train/test-combinations is controlled (as reported on the x-axis in Figure 4) and how this relates to the parameter "r" that ensures that each attribute value is observed at least "r" times. It would be great if you could clarify this.

* There are a number of references to tables / figures broken, which made it difficult to read (eg. line 205 should be figure 7 in the appendix) and there are many more.

* The importance of incorporating modularity in the encoder is in my view intimately related to the binding problem in neural networks and the "superposition catastrophe" when superimposing information about multiple objects using distributed representations. A recent survey on this topic by Greff et al. (2020), see problem of segregation, may be used to draw a connection and strengthen this argument further.

* The variation with batch norm explored in the appendix seems an arbitrary choice. It would be helpful to motivate why you are exploring this particular variation.

* The terms "systematic generalization" and "out of distribution" are interchanged in the text without pointing out their connection.

* It would be great to be consistent about the number of seeds across the paper and increase the number of seeds for Figure 4 from 2 to 5.

* There are a number of naming conventions that currently hamper readability. For example, the "Find module" and "Residual module" that are not very descriptive of the broader concept. Inputs such as s_x and s_y. variations such as "Condition A" and "Condition B", etc.

* Please take a moment to go through the references to correctly cite prior work that has been published (as opposed to citing the arxiv version)

**References**

Greff, K., van Steenkiste, S., & Schmidhuber, J. (2020). On the binding problem in artificial neural networks. arXiv preprint arXiv:2012.05208.

**Post-rebuttal Update**

I remain with my current score of a reject (4). The reasons for this I have outlined in detail in a public reply above.


**Time Spent Reviewing:**

9

---

### Official Review · Reviewer_BWzo · 2021-07-16

**Rating:** 6
**Confidence:** 3

**Summary:**

This paper investigates the effect of the degree of modularity on the systematic generalization of neural module networks (NMNs). Specifically, the authors study how the degree of separation or sharing of the weights across sub-tasks at the image encoder, intermediate modules, and classifier levels affect out-of-distribution generalization on multi-attribute MNIST, SQOOP, and CLEVR-CoGenT with systematic test splits. The sub-tasks are either grouped under a single group called "all", or in groups of high-level functional classes, or in groups of 1. Weights within a group are shared, but the module operation depends on a sub-task embedding (this is done through straightforward architectures in the extremes of modularity). The authors explore four combinations of modularity settings across image encoder, intermediate modules, and classifiers. In experiments with multi-attribute MNITST and CLEVR-CoGenT, the authors show the benefit of separating groups in the image encoder. For SQOOP, they show an advantage of using sub-task-level intermediate modules.

**Limitations And Societal Impact:**

The authors adequately discuss the limitations of having manually tuned the degree of modularity, and that of using ground-truth layouts.

**Main Review:**

This is an interesting paper that provides insights into desirable architectures for modules in NMNs that can systematically generalize. The paper presents new studies of the effect of the degree of modularity in neural module networks. The authors also propose adding modularity in the image encoder as a novel contribution, which significantly impacts the generalization for the chosen datasets. They also show a benefit of an intermediate level of modularity for the intermediate modules.

Main concern: For systematic generalization in module networks, a desirable property is that each module learns to perform a distinct sub-task, independent of the program layout. In this way, each module is a distinct computation block, and modules can be arranged arbitrarily in any legal order to represent a valid computation graph that performs the desired computation. For this goal, I appreciate the authors using ground-truth program layouts to focus on the ability of the modules to learn such behavior without the added noise from the program generator. However, a good test for this type of systematicity is evaluating with systematically OOD test program layouts, to see if the learned modules can simply be organized in different layouts to still achieve correct predictions. By focusing more on OOD image distributions, the cause for better generalization is a bit less clear. The authors change the image distribution for both multi-attribute MNIST and CLEVR by using CoGenT. For instance, between CLEVR-CoGenT and CLOSURE, I believe CLOSURE is a better test of systematic generalization than CLEVR-CoGenT for this study since it considers novel module arrangements (because the question distribution changes) for the same image distribution. I do see that the authors included CLOSURE results in the appendix, but the proposed method does not provide benefits in that case. In SQOOP, the bias in the question setting does not provide any useful signal since both models achieve near-perfect systematic generalization. I think it is fine not to beat existing baselines as long as there is something to learn, but the experiments for these settings are minimal.

Other concerns/questions:
- With the three types of groupings presented in the paper, along with three levels of the pipeline, there could be a total of $3^3=27$ possible combinations of modularity at different levels. Did the authors experiment with all of them? If not, how did they choose the specific subset to experiment with?
- Lines 78-79: Is modularity only imposed on the classifier in the presence of a binary classification task? What happens in the case of CLEVR, where it is not a binary classification task? For the one network per sub-task setting, is each sub-task classifier a binary classifier for all datasets, including CLEVR?
- Why is Section 5 a separate section instead of just another result in Section 4?

Other notes:
- The paper lacks a related work section to set up the context in the literature thoroughly. But it does provide many of the important references in the paper body.
- Lines 25-27: "et al." does not need to be italicized. Also, it is synonymous with "and others", so "X et al." are multiple people. Thus, it would be best if you said "X et al. have shown..." instead of "X et al. has shown..." (same with "report" vs "reports" in line 27).
Lines 65-68: It is a bit difficult at first to follow the authors' introduction of the terms "sub-task" and "group", how they relate to "blocks". It would be great to illustrate these terms with a short example program.

The results of the paper are significant and important to the community. However, it is difficult to take away the message clearly from the limited presented experiments that test the compositionality of the modules in the in-distribution image, out-of-distribution layout setting.

**Time Spent Reviewing:**

9

---

### Author Response · Authors · 2021-08-12
**To all: Thanks for the insightful comments**

We thank the reviewers for their feedback, which is helping us to improve the paper. The discovery introduced in the paper is that the degree of specialization of NMN modules leads to substantial gains of systematic generalization. This is an important finding to overcome bias in ML, which is arguably the most urgent problem to be addressed in ML nowadays. However, the paper did not excite any of the reviewers to our expectations, and we are going to address this by improving the manuscript taking their comments into account.

We want to point out that the criticisms of the negative review stems from a lack of understanding of the field---the review main point is that our discovery is not surprising because it is obvious that large gains of **systematic generalization** can be achieved by training modules to do less complex functions, even with less training data (sic). No references were provided.

The rest of reviewers suggest accepting the paper but share the concern that we did not try all the possible combinations of networks and/or datasets. While we have tried an extensive amount of networks and datasets (please take the appendices into account), we agree that trying more of them is always desirable. Note also that the majority of published works in this field focus on alleviating the effect of only one type of bias, and in our paper we alleviate the effect of bias in the combination of object attributes. In terms of analyzing network variations, reviewers mention that we should report architectures that have specialized modules in one stage and in the next stage have no specialization, ie. architectures that revert the specialization of the modules in previous stages. It is always interesting to know results from all kinds of network architectures, but in this case results are not relevant for the main claim of the paper.

Finally, we would like to thank once more the reviewers for providing their candid feedback, as it is helping us to improve the paper.  All their points are going to be included and clarified in the final version of the paper.

---

### Author Response · Authors · 2021-08-27
**Additional experiments suggested by reviewers strengthen the main claim of the paper**

We got curious, as the reviewers are, to know more about architectures that have specialized modules in one stage and in the next stage have no specialization. Therefore we executed additional experiments on NMNs with “group - all - all” and “sub-task - all - all” architectures, on the four datasets of the Multi-attribute MNIST family. See tables below (recall that “r” denotes the data variability, ie. the minimum amount of times each attribute instance is included in the training combinations). The first three columns are the configurations already reported in the paper, and the last two columns are the new configurations. We can observe that results further strengthen the main claim of the paper (which is, quoting from the abstract, “tuning the degree of modularity in the network, especially at the image encoder stage, facilitates substantial improvements of systematic generalization”).

Many thanks to the reviewers for suggesting these experiments!



***Attribute extraction from single object (across two trials)***

| r | **(all - all - all)** | (group - group -group) | (sub-task - sub-task - sub-task) | **(group - all - all)** | **(sub-task - all - all)** |
|-------------|:-------------:|:-------------:|:-------------:|:-------------:|:-------------:|
| ***1*** | 62.92 ± 1.63 | **67.01 ± 3.08** | 63.75 ± 3.33 | 64.71 ± 2.68 | 64.28 ± 2.89 |
| ***2*** | 69.22 ± 0.36 | 73.87 ± 0.44 | 73.73 ± 0.28 | **74.60 ± 0.90** | 73.96 ± 0.098 |
|  ***5*** | 76.44 ± 1.48 | 85.42 ± 0.72 | 85.97 ± 1.05 | **89.80 ± 0.21** | 85.44 ± 0.99 |
| ***8*** | 85.65 ± 1.39 | **99.01 ± 0.20** | 94.24 ± 0.79 | **99.02 ± 0.12** | 92.71 ± 0.32 |
| ***10*** |87.93 ± 4.78| **99.35 ± 0.15** | 95.56 ± 0.33 |  **99.15 ± 0.052** | 94.77 ± 0.57 |
| ***20*** | 97.76 ± 0.40|**99.58 ± 0.023** |**99.33 ± 0.052** | **99.62 ± 0.077** | **99.37 ± 0.067**|

***Attribute extraction from multiple objects (across two trials)***

| r | **(all - all - all)** | (group - group -group) | (sub-task - sub-task - sub-task) | **(group - all - all)** | **(sub-task - all - all)** |
|-------------|:-------------:|:-------------:|:-------------:|:-------------:|:-------------:|
| ***1*** |  60.40 ± 2.58 | 63.46 ± 0.47 | 62.87 ± 1.83 | **66.48 ± 0.42** | 63.35 ± 1.74 |
| ***2*** | 66.96 ± 2.16 | **73.96 ± 1.88** | 70.00 ± 1.69 | 72.97 ± 1.80 | 70.29 ± 1.29 |
|  ***5*** |  75.64 ± 0.40 | 83.76 ± 1.75 | 83.65 ± 0.37 | **87.20 ± 0.69** | 84.34 ± 0.48 |
| ***8*** | 80.23 ± 1.31 | **97.40 ± 0.82** | 91.58 ± 0.73 | 96.62 ± 0.55 | 92.36 ± 0.70 |
| ***10*** | 82.83 ± 0.89 | **98.43 ±0.029**| 94.88 ± 0.66 | **98.83 ± 0.042** | 93.97 ± 0.24 |
| ***20*** | 95.32 ± 0.53 | 98.27 ± 1.21 | **99.29 ± 0.01** | **99.61 ± 0.052** | **99.03 ± 0.19** |

***Attribute comparison between pairs of separated objects (one trial)***

| r | **(all - all - all)** | (group - group -group) | (sub-task - sub-task - sub-task) | **(group - all - all)** | **(sub-task - all - all)** |
|-------------|:-------------:|:-------------:|:-------------:|:-------------:|:-------------:|
| ***1*** | 68.5833 | 76.5262 | 75.2190 | 72.6119 | **77.1500** |
| ***2*** | 81.1738 | 84.6667 | **86.3714** | 84.5405 | 84.7357 |
| ***5*** | 95.7071 | 93.4548 | 94.4333 | **96.4405** | 95.7262 |
| ***8*** | **98.0286** | **98.0905** | **98.7452** | **98.7571** | **98.3238** |
| ***10*** | **99.0571**| **99.3619** | **99.0429** | **99.1738** | **99.2024** |
| ***20*** | **99.3238** | **99.5619** | **99.6929** | **99.3071** | **99.5905** |

***Comparison between spatial positions (one trial)***

| r | **(all - all - all)** | (group - group -group) | (sub-task - sub-task - sub-task) | **(group - all - all)** | **(sub-task - all - all)** |
|-------------|:-------------:|:-------------:|:-------------:|:-------------:|:-------------:|
| ***1*** | 67.5381 | 68.1690 | **70.0881** | **70.4333** | 67.7286 |
| ***2*** | 81.5095 | 81.0881 | **83.2000** | **83.4357** | 80.9810 |
| ***5*** | 89.0000 | **97.1000** | 94.1619 | 96.6452 | 92.6667 |
| ***8*** | 93.9286 | **97.8952** | 96.9071 | **97.7524** | 95.3524 |
| ***10*** | 94.8048 | **97.9071** | **97.4881** | **97.8381** | 96.3143 |
| ***20*** | 97.9000 | 97.7643 | **98.1524** | **98.1095** | 97.8286 |


Although Trial 2 for “attribute comparison between pairs of separated objects” and “comparison between spatial position” are still ongoing, we do not expect a large difference in performance, given the main trend across different tasks. We are adding these results in the final manuscript as well.

---

### Decision · Program_Chairs · 2021-09-27

**Decision:**

Accept (Poster)

**Comment:**

The paper presents a careful investigation into how modularity affects systematic generalization across a number of synthetic datasets. The key contribution is to highlight the role of modularity, especially when introduced at the early stage of the network.

The most important concern raised by reviewers is that the experimental setting is limited. Neural Module Network, the network that all experiments are based on, is very rarely used in practice, and all the datasets are synthetic. On top of this, ground truth grouping or program layout is provided to the network, which wouldn't be normally available. It is genuinely difficult to tell how the results will translate into performance improvements more broadly.

On the positive side, the paper is technically sound and the conclusions are interesting, as agreed by most reviewers. With the added experiments in the rebuttal, the experimental data clearly supports that adding modularity to NMNs improves systematic generalization. Based on this, I am happy to support acceptance of the work. I would like to strongly encourage the Authors to broaden their experimental setting in future work, so that these results have real impact on how we train neural networks. Please remember to address all reviewers' comments in the camera ready version.